# Ability of the ash dieback pathogen to reproduce and to induce damage on its host are controlled by different environmental parameters

**Benoit Marçais** [1]*, **Arnaud Giraudel**[1], **Claude Husson**[2]

**1** Université de Lorraine, INRAE-Grand-Est, UMR1136 Interactions Arbres/Microorganismes, Nancy, France, **2** Département de la Santé des Forêts, Ministère de l'Agriculture et de la Souveraineté Alimentaire–DGAL, Paris, France

* benoit.marcais@inrae.fr

**Data Availability Statement:** Data are available in DRYAD repository (doi:10.5061/dryad. kh189328x).

## Abstract

Ash dieback, induced by an invasive ascomycete, *Hymenoscyphus fraxineus*, has emerged in the late 1990s as a severe disease threatening ash populations in Europe. Future prospects for ash are improved by the existence of individuals with natural genetic resistance or tolerance to the disease and by limited disease impact in many environmental conditions where ash is common. Nevertheless, it was suggested that, even in those conditions, ash trees are infected and enable pathogen transmission. We studied the influence of climate and local environment on the ability of *H. fraxineus* to infect, be transmitted and cause damage on its host. We showed that healthy carriers, i.e. individuals showing no dieback but carrying *H. fraxineus*, exist and may play a significant role in ash dieback epidemiology. The environment strongly influenced *H. fraxineus* with different parameters being important depending on the life cycle stage. The ability of *H. fraxineus* to establish on ash leaves and to reproduce on the leaf debris in the litter (rachises) mainly depended on total precipitation in July-August and was not influenced by local tree cover. By contrast, damage to the host, and in particular shoot mortality was significantly reduced by high summer temperature in July-August and by high autumn average temperature. As a consequence, in many situations, ash trees are infected and enable *H. fraxineus* transmission while showing limited or even no damage. We also observed a decreasing trend of severity (leaf necrosis and shoot mortality probability) with the time of disease presence in a plot that could be significant for the future of ash dieback.

## Author summary

Ash dieback is an important tree disease induced by *Hymenoscyphus fraxineus*, an invasive fungus. We studied the influence of climate and local environment on the ability of *H. fraxineus* to infect, be transmitted and cause damage on its host. We showed that healthy carriers, i.e. individuals showing no dieback but carrying *H. fraxineus*, exist and may play

**Funding:** BM. received grant (2014-18) from Forest Health Department, French Ministry of Agriculture and Forestry https://agriculture.gouv.fr/le-departement-de-la-sante-des-forets-role-et-missions The funder participated in the data collection, in particular for plots in Brittany and Charente. The funder had no role in study design, analysis, decision to publish, or preparation of the manuscript BM. received grant (2018-21) from H2020 HOMED. https://homed-project.eu/ The funder had no role in study design, data collection and analysis, decision to publish, or preparation of the manuscript. BM. received grant (2021-22). Overseen by the French National Research Agency (ANR) as part of the "Investissements d'Avenir" program (ANR-11-LABX-0002-01, Lab of Excellence ARBRE) https://mycor.nancy.inra.fr/ARBRE/ The funder had no role in study design, data collection and analysis, decision to publish, or preparation of the manuscript.

**Competing interests:** The authors have declared that no competing interests exist.

a significant role in ash dieback epidemiology. While the ability of *H. fraxineus* to establish and to reproduce on leaves mainly depended on summer rainfall, the damage it causes to the host depends on other parameters (high summer temperature in July-August, tree cover). As a consequence, in many situations, ash trees are infected and enable *H. fraxineus* transmission while showing limited or even no damage. This is in particular the case of non-forest situations such as hedges and isolated trees.

## Introduction

The existence of healthy carriers, i.e. infected individual that remain asymptomatic, is not recognized as an important feature in plant epidemiology [1]. The existence of healthy carriers has especially been reported for weak pathogens or those that can induce damage only on stressed hosts (so-called latent pathogens, [2]). This is known to be a significant feature for important stress-induced tree diseases such as *Diplodia* shoot blight, sooty bark disease of maples or *Botryosphaeria* cankers [3,4,5]. However, high production of inoculum usually occurs only at symptom onset. By contrast, the existence of healthy carriers is seldom reported for primary pathogens that do not need host stress to induce significant host damages. Tolerance has been reported as a mechanism used by plants to cope with infection [6]. This mechanism postulates that some individuals will limit the impact of infection on their fitness while sustaining significant pathogen load and enabling pathogen multiplication. However, the presence of individual plants that provide important levels of inoculum while exhibiting limited or even no visible symptoms is seldom reported. Disease severity is usually assumed to correlate positively with the multiplication of the pathogen within the host [7]. Nevertheless, at a different level, when comparing different species, it has been shown that poor correlation may exist between vulnerability (reduction of fitness caused by the infection) and competence (ability to sustain the epidemic by providing large amounts of inoculum). A striking example is given by Sudden Oak Death that developed in California in the last decades: while this disease, induced by *Phytophthora ramorum*, affects primarily *Fagaceae*, killing *Quercus* and *Lithocarpus* species, the main inoculum producer is a laurel, *Umbellularia californica* which coexists in natural forest and does not appear to suffer much fitness reduction (so-called reservoir host, [8,9,10]).

Such a discrepancy between a pathogen's ability to establish in a stand and to produce inoculum and its ability to induce damages is an important difficulty for assessing the climatic niche of a pathogen. This assessment is a critical step to evaluate the risk associated with pathogen occurrence [11]. The most widespread approach to develop species distribution models (SDM) is to relate the species known occurrence to environmental parameters, in particular climate, through statistical analyses [12]. If large discrepancies exist between a pathogen presence and its ability to cause serious damage, this will lead to important under-reporting and to poor available data. By contrast, mechanistic SDM, which relies on the assessment of environment influence on the different stages of the pathogen cycle [13] could be a suitable alternative to assess the climatic niche of a pathogen in such conditions.

*Hymenoscyphus fraxineus* is an invasive pathogen that has caused ash dieback throughout Europe since the late nineties and has jeopardized *Fraxinus excelsior* stands [14]. *H. fraxineus* fulfils its life cycle on ash leaves. Those are infected by ascospores during late spring and summer and remain asymptomatic until late August when necrotic leaf lesions develop [15]. *H. fraxineus* then colonises the leaf petiole and may infect young shoots, inducing dieback [14]. Most of the inoculum production occurs on leaf debris in the forest litter (so-called rachis that encompasses the petiole and the main vein of the composite leave, on which fruit bodies i.e.

apothecia develop, [14]). By contrast, very limited inoculum production occurs on bark of diseased shoots which is considered to represent a dead-end for *H. fraxineus* [16]. It has been suggested that, in some conditions, *H. fraxineus* may reproduce on ash leaves without inducing significant dieback. Some *F. excelsior* individuals are not affected by *H. fraxineus*, even under high inoculum pressure [17]. These individuals do not sustain extensive lesions when artificially inoculated at the shoot level [18]. It was then suggested that their fallen leaves could nevertheless sustain the pathogen reproduction and that the mechanism involved in containing the disease might be tolerance instead of resistance [16]. However, no experimental data is yet available to sustain this claim. Another reported tolerance mechanism would be early abscission that could prevent transfer of the pathogen from infected leaves to shoots but still allow sexual reproduction on fallen rachises [17,19,20]. Also, environmental conditions may influence differently dieback and colonisation of the rachises in the litter. Grosdidier et al. [21] showed that in open canopy conditions, the infection levels of ash rachis by *H. fraxineus* in the litter was similar compared to forest stands with a close canopy despite much lower levels of crown dieback. Those data would suggest that healthy carriers could exist in the *F. excelsior* population, with individuals able to be infected in specific conditions at the leaf level and thus able to produce inoculum while sustaining very limited to no dieback.

We thus examined whether ash trees could in some conditions behave as healthy carriers. More specifically, we tested if different environmental variables may favour important stages of ash dieback, namely the ability to colonise leaf rachises and reproduce on them and the ability to induce damage on ash trees, either leaf necrosis at the end of the summer or subsequently shoots mortality. We propose that ash behaving as healthy carriers could occur in locations where the environment promotes infection and reproduction of *H. fraxineus* but not symptom development.

## Results

Data were available for 55 plot*year combinations for leaf necrosis frequency (Fig 1, S1 Table). Less data were available for shoot mortality (44 site*year combinations) because some plots were lost during winter (logging, 4 plots) or were not rated in spring (S1 Table). Altogether, *H. fraxineus* was detected by the qPCR assay in 86% of the 63 tested leaves with necrosis. However, the result depended on the plot and year, with lower detection frequency in plots with very little leaf necrosis (75% of tested leaves with necrosis when overall leaf necrosis frequency was lower than 5%; 84% when it was between 5 and 15%; and 95% when it was higher than 15%).

### Ability of *H. fraxineus* to complete its cycle on ash saplings with different dieback severity

This section refers to the specific study done in two of the plots, Gremecey and Seichamps in 2014–15. Most of leaf symptoms that were observed in early September 2014 were partial leaf necrosis with one or several leaflets with necrosis but most leaflets still green. No shoot mortality was observed in September 2014, the mortality occurring later, during the winter. The probability of leaf necrosis was significantly higher in Gremecey than in Seichamps (likelihood Chi-square = 4.55, p = 0.03, 38.0 ± 6.5% in Gremecey and 20.6 ± 4.4% in Seichamps). The percentage of leaves with necrosis was similar on saplings showing either no or severe dieback (dieback score of 0–1 versus 3, likelihood Chi-square of 0.7084, p = 0.40, Fig 2A). This was observed in the two studied plots as the plot * dieback status interaction had no significant effect on leaf necrosis frequency (likelihood Chi-square = 1.32, p = 0.25).

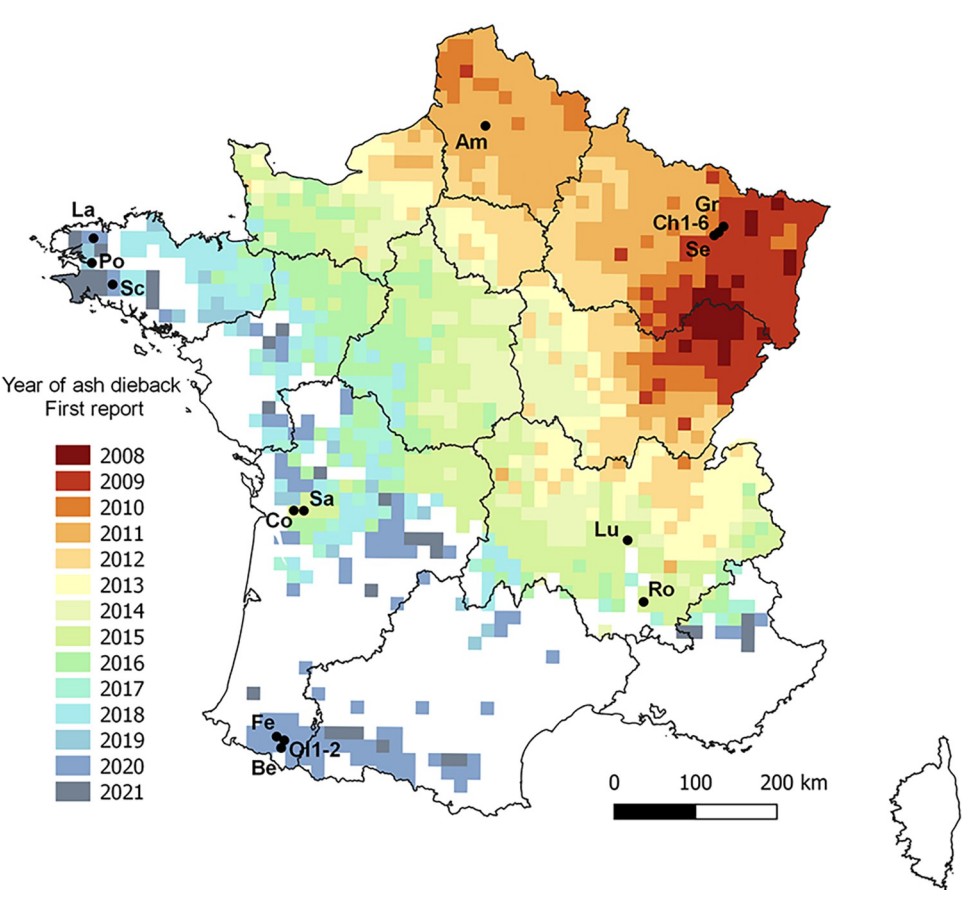

**Fig 1. Location of the studied ash stands (black dot) relative to the year of first ash dieback report.** Data from the Département de la Santé des Forêts. Am, Fréchencourt, Be, Sarrance, Ch1-6, Champenoux (6 plots), Co, Colombier, Fe, Ance-Féas, Gr, Gremeçey, La, Landivisiau Lu, Lupé, Ol1-2, Oloron-Sainte-Marie (2 plots), Po, Pont-de-Buis, Ro, Roche-sur-Grane, Sa, Salignac-sur-Charente Sc, Scaër, Se, Seichamps). The region border shapefile can be uploaded at https://geoservices.ign.fr/telechargement.

*H. fraxineus* was able to produce apothecia on rachis of both the saplings with no and severe dieback at the same rate (crown status of 0–1 versus at least 3, likelihood Chi-square = 1.19, p = 0.28, Fig 2B). The saplings with no dieback experienced limited mortality of the marked shoot while the shoot mortality was high for saplings showing dieback (likelihood Chi-square = 6.99, p = 0.01, Fig 2C). In summer 2015, the crown status of the saplings had not changed, i.e. only very minor signs of *H. fraxineus* infection were observed on saplings with no dieback symptoms in 2014.

## Relationship between leaf necrosis in fall and subsequent shoot mortality

Shoot mortality in spring was strongly related to the frequency of leaves with necrosis at the end of the previous summer (Fig 3A and 3B, Table 1). It is noteworthy that a small proportion of shoots that showed no leaf necrosis in September were nevertheless dead in the next spring (Fig 3A). Shoot mortality was very different on ashes with different dieback status (Table 1, Fig 3A). The odds ratio for shoot mortality of an ash with dieback versus one without was 4.1 (95% credible interval CI [2.4, 7.0]). By contrast, leaf necrosis did not significantly depend on tree dieback status (Table 1, mean leaf necrosis frequency of 12.9 ± 3.8% for ashes without dieback and 16.3 ± 6.3% for ashes with dieback). Both leaf necrosis and shoot mortality were

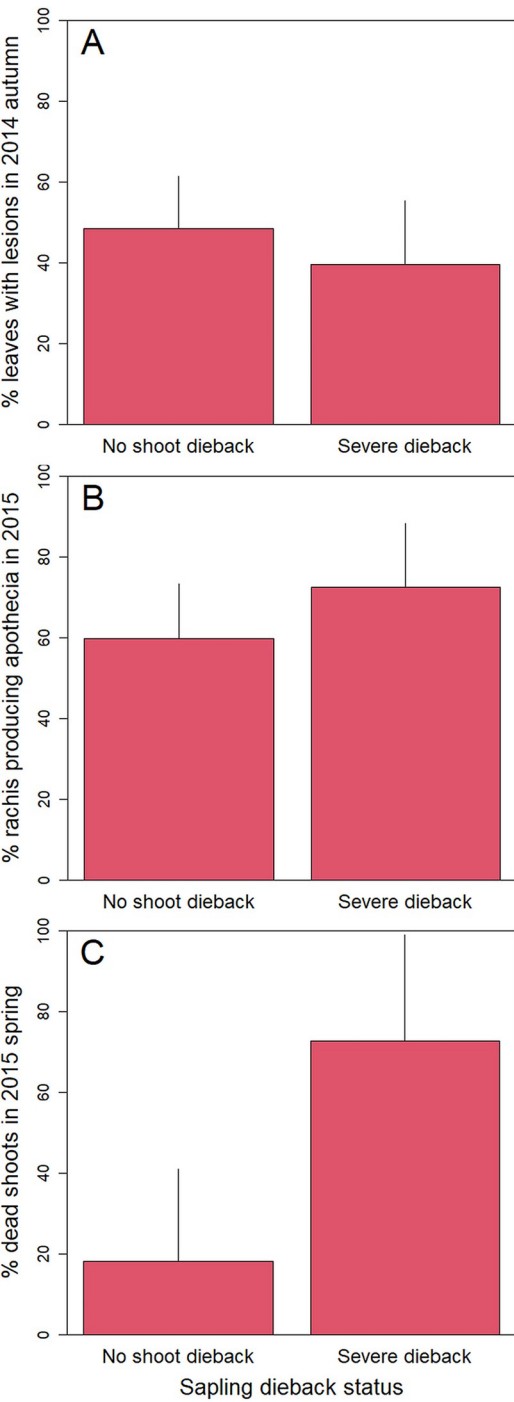

**Fig 2. Impact and ability of *H. fraxineus* to complete its cycle on ash saplings with either no or severe dieback.** (A) % leaves with necrosis in autumn 2014 (likelihood Chi-square of 0.7084, p = 0.40). (B) % of rachis length with presence of *H. fraxineus* apothecia in April 2015 (likelihood Chi-square = 1.19, p = 0.28). (C) % of rated shoot with bark infection in summer 2015 (likelihood Chi-square = 6.99, p = 0.01).

strongly affected by the tree random factor with a similar range of standard deviation (Table 1). This means that, even after other variables of the model are taken into account (% tree cover, dieback status), some of the rated ashes consistently showed more leaf necrosis or shoot mortality across years. This effect was of the same magnitude for leaf necrosis and for shoot mortality.

Shoot mortality was also significantly increased at higher tree cover (Table 1, Fig 3B), with an odds ratio of 1.2 (95% CI [1.0, 1.3]) per 10% tree cover. By contrast, leaf necrosis was not affected by tree cover (Table 1). The environment random factor induced a level of variability similar for leaf necrosis and shoot mortality (Table 1). As no meso-climate effect was included in this model, high environment random factors were expected.

Early defoliation was significantly associated with shoot mortality, with shoot mortality increasing with the defoliation level (Table 1). Defoliation appeared to be a consequence of infection by *H. fraxineus* and significantly increased with leaf necrosis probability (likelihood Chisq of 58.3, p< 0.0001).

**Relationship between leaf necrosis or shoot mortality and climate.** The preliminary analysis showed that the best fit for foliar necrosis occurred for climatic conditions in June although the difference was minor (DIC of 99.8 compared to DIC of 100.9 for 15 May to 15 June and 113.7 for 15 June to 15 July). Likewise, for shoot mortality, the best fit occurred for climatic conditions in September to end of December (DIC of 99.8 compared to DIC of 103.5 for climatic conditions in September-October).

The model fitted adequately the data, with a proper residual distribution and an acceptable prediction of observed values in the cross-validation (S1 Fig). The main parameter influencing leaf necrosis was TX78, the average of daily maximal temperature in July and August with a strong negative effect of high summer temperatures (Table 2, Fig 4A, odds ratio of 0.70, 95% CI [0.63, 0.84]). High average temperature in June increased the probability of leaf necrosis (Table 2, odds ratio of 1.33, CI [1.11, 1.52]). By contrast, rainfall either in June or in July-

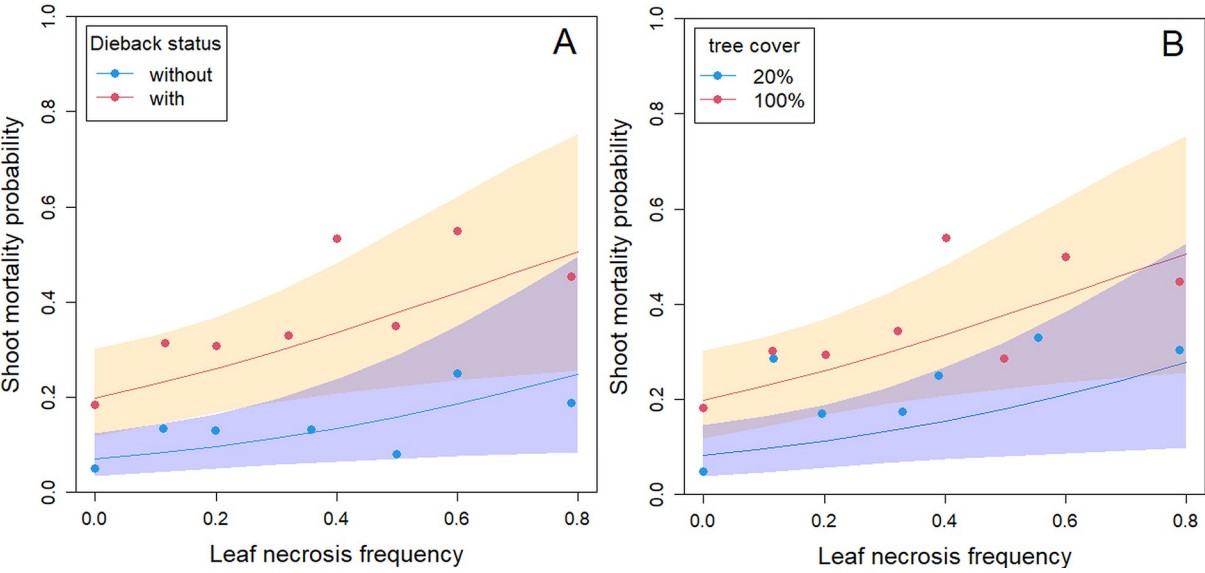

**Fig 3. Modelled and observed shoot mortality for ash trees of different level of dieback.** (A) Site with different dieback status (with dieback, score 0–1 or without, score of 2–3). (B) Sites with different tree cover. Points represent the observed data, pooled by increasing 10% leaf necrosis frequency. For tree cover, blue dots represent sites with 9–25 tree cover while red dots represent sites with 75–100% tree cover. Points representing less than 10 shoots are not represented.

**Table 1. Relationship between leaf necrosis and shoot mortality at the shoot level.** Parameters of the hierarchical Bayesian model.

| Variable | Leaf necrosis [95% credible interval] | Shoot mortality [95% credible interval] |
|---|---|---|
| Sapling dieback status (DS) | 0.27 [-0.01, 0.55] | **1.27 [0.82, 1.75]** |
| Site tree cover (TC) | -0.01 [-1.14, 1.16] | **1.40 [0.07, 2.23]** |
| Defoliation at September rating (%,DEF) | - | **1.53 [0.38, 2.68]** |
| Probability of necrotic leaves (pl) | - | **1.75 [0.24, 3.27]** |
| Tree random factor (sd, Tree.l and Tree.s) | 0.93 [0.82, 1.06] | 0.89 [0.55, 1.24] |
| Site*year random factor (sd, Env.l and Env.s) | 0.95 [0.65, 1.32] | 1.22 [0.97, 1.54] |
| Deviance | 4615 [4532, 4698] | |

NOTE. Parameter with a 95% credible interval that does not contain 0 are considered significant and are marked in bold. Positive values indicate an association between high levels of both the symptom and of the environmental variable.

August was not related to leaf necrosis probability. Lastly, we observed a trend of decreasing leaf necrosis probability with the time of ash dieback in the area (Table 2, Fig 4C, odds ratio of 0.88, CI [0.83, 0.93]).

The shoot mortality was mainly linked to the average temperature in autumn, with mortality probability decreasing with increasing autumn temperatures (September to December, Table 2, Fig 4D, odds ratio of 0.69, CI [0.54, 0.86]). Rainfall in autumn was not related to shoot mortality (Table 2). As in the shoot level model, shoot mortality increased significantly with leaf necrosis frequency (Table 2). As a consequence, all parameters influencing leaf necrosis probability also influenced shoot mortality (Fig 4E, 4F and 4G).

**Table 2. Relationship between leaf necrosis or shoot mortality and climate parameter (plot level).** Parameters of the hierarchical Bayesian model.

| Variable | Symptom | Parameter [95% credible interval] |
|---|---|---|
| Rain June (RA6, $\alpha_2$) | Leaf necrosis | 0.04 [-0.16, 0.25] |
| Temperature June (TM6, $\alpha_3$) | Leaf necrosis | **0.69 [0.37, 0.99]** |
| Rain Junly-August (RA78 ($\alpha_4$) | Leaf necrosis | -0.20 [-0.52, 0.11] |
| Average daily maximal temperature July-August (TX78, $\alpha 5$) | Leaf necrosis | **-1.11 [-1.50, -0.69]** |
| N year ash dieback presence (LDP, $\alpha 6$) | Leaf necrosis | **-0.37 [-0.58, -0.17]** |
| Site random factor (Site.l, sd) | Leaf necrosis | 0.21 [0.01, 0.49] |
| Rain September-December (RA912, $\beta_2$) | Shoot mortality | -0.06 [-0.43, 0.29] |
| Temperature September-December (TM912, $\beta_3$) | Shoot mortality | **-0.83 [-1.30, -0.39]** |
| Site tree cover (TC, $\beta_4$) | Shoot mortality | **0.77 [0.07, 1.52]** |
| Probability of necrotic leaves (pl, $\beta_5$) | Shoot mortality | **4.55 [4.38, 8.12]** |
| Site random factor (Site.s, sd) | Shoot mortality | 1.21 [0.68, 2.01] |
| Deviance | | 3745 [3667, 3826] |

NOTE. Parameter with a 95% credible interval that does not contain 0 are considered significant and are marked in bold. The variables are standardized to have zero mean and one standard deviation with the exception of pl, the leaf necrosis likelihood. Positive values indicate an association between high levels of both the symptom and of the environmental variable.

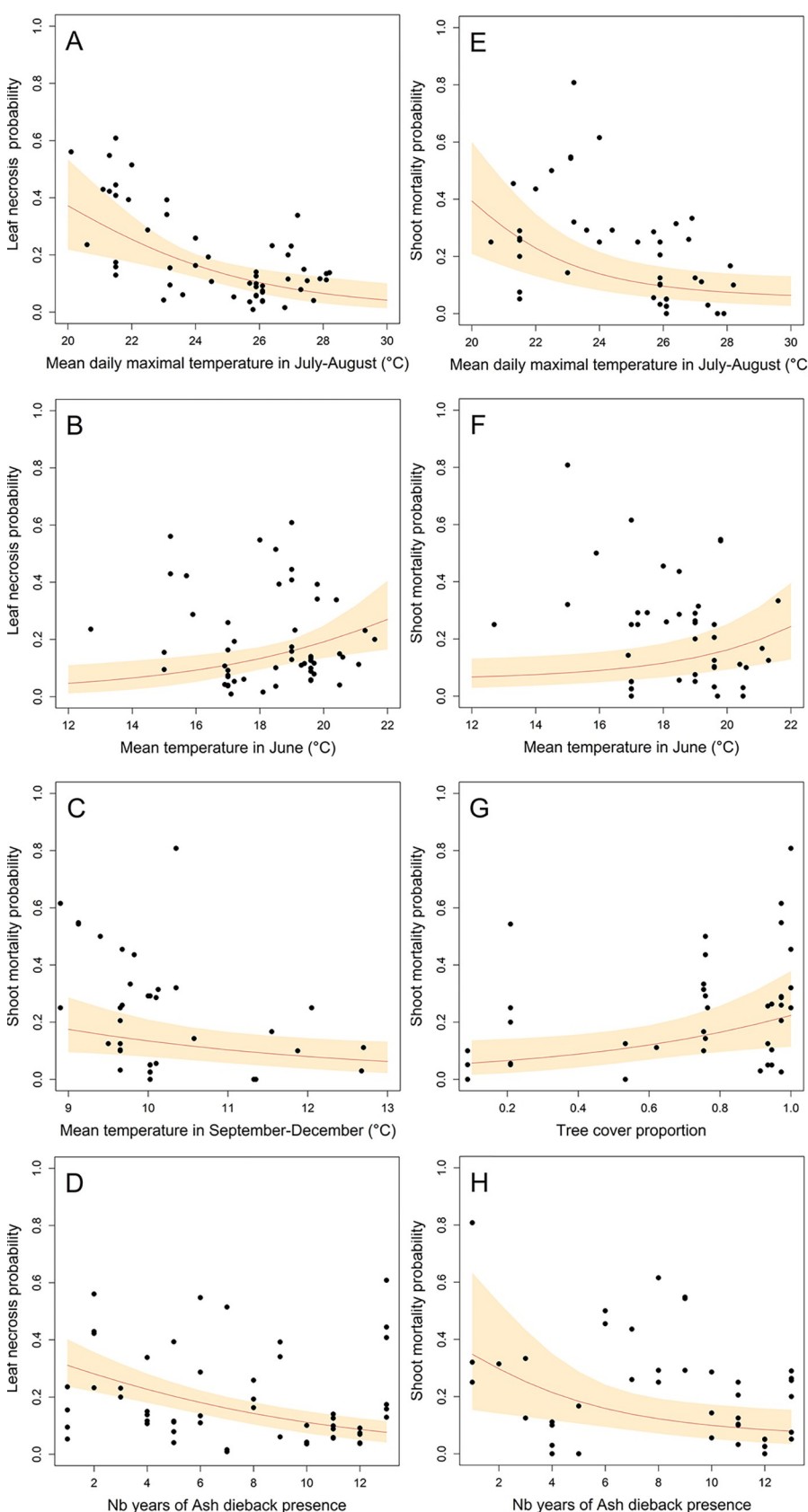

**Fig 4. Modelled and observed leaf necrosis and shoot mortality depending on climatic parameters and sites conditions.** The curves represent the predicted response for a given parameter, all other parameters being fixed at the average value, except for the number of year of ash dieback presence that is set to 1. The orange area (grey in printed version) represent the 95% confidence interval. Dots represent oberved values.

## Ability of *H. fraxineus* to produce apothecia on overwintered rachises in the fructification assay

The presence of *H. fraxineus* in rachises at leaf fall was tested in 2020 and 2021 in part of the sites by qPCR. It showed striking differences between years and locations (Table 3). Moreover, it was not significantly linked to either the leaf necrosis frequency measured in September, two months before (Pearson R = 0.21, p = 0.44) or the subsequent shoot mortality (Pearson R = 0.266, pe = 0.32). In Champenoux, although the frequency of leaf necrosis was low both in 2019 and 2020, the proportion of rachis in which *H. fraxineus* was detected by qPCR was very different between the two years (Table 3). Another striking feature was that the frequency of *H. fraxineus* detection by qPCR in rachises at leaf fall was almost ten-fold higher than the frequency of necrotic leaves.

The proportion of rachises producing apothecia in the fructification assay in laboratory conditions showed very high variability, with a range of 0.01 to 0.99. This strongly depended on the year with mean values of $0.26 \pm 0.11$ in 2016, $0.52 \pm 0.17$ in 2017, $0.79 \pm 0.09$ in 2019 and $0.37 \pm 0.23$ in 2020. The proportion of rachises producing apothecia was significantly correlated to the proportion of rachises in which *H. fraxineus* was detected by qPCR at leaf fall (Pearson R = 0.71, pvalue< 0.01).

The proportion of rachises producing apothecia in the fructification assay was not significantly related to the frequency of leaf necrosis in the previous September rating (Fig 5A). It was not significantly related to the tested climatic parameters during the overwintering period (Table 4). By contrast, it was significantly related to the rainfall during the previous summer (sum of rainfall in July-August, Table 4, Fig 5B). It was also not related to the % of tree cover (Table 4).

## Climate suitability in France for the different stage of *H. fraxineus* life cycle

Maps were produced using the developed models (Tables 2 and 4) for an average site (site random factor fixed at 0) and for years 2010 to 2020. *H. fraxineus* should be able to colonise leaves, be present on rachises shed in autumn and be able to produce apothecia in spring in most of France (Fig 6A). By contrast, its ability to produce damage should be much more limited, in particular in southern France (Fig 6B–6D). However, its ability to cause damage to ash trees is strongly dependent on local site conditions: comparison of Fig 6C and Fig 6D show large difference of predicted shoot mortality depending on the % tree cover. Uncertainty in parameter estimation very significantly affected both the maps of necrotic leaves / shoot mortality probability (S2 Fig).

**Table 3. Colonisation of rachises at leaf fall in November.**

| Area | Year | Frequency of necrotic leaves in September | Frequency of *H. fraxineus* detection on rachises in November (qPCR) |
|---|---|---|---|
| Champenoux | 2019 | $0.09 \pm 0.03$ | $0.87 \pm 0.19$ |
| | 2020 | $0.05 \pm 0.02$ | $0.35 \pm 0.12$ |
| Pyrenean piedmont | 2020 | $0.12 \pm 0.05$ | $0.60 \pm 0.18$ |

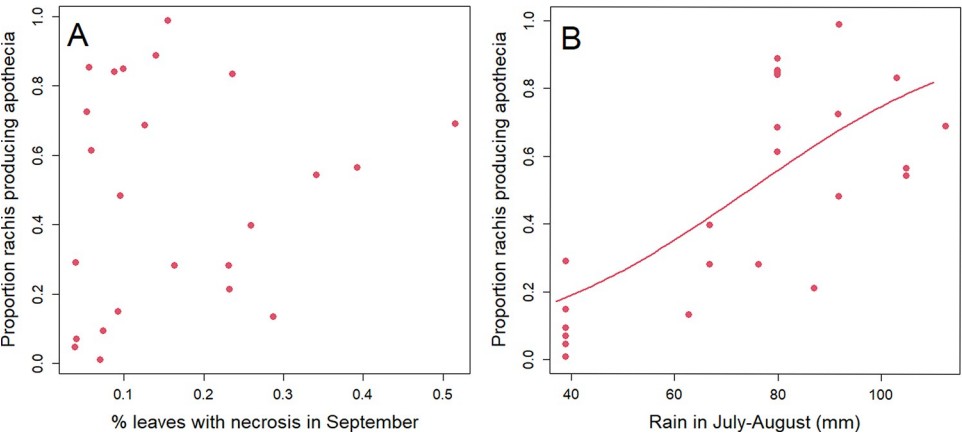

**Fig 5. Ability of *H. fraxineus* to produce apothecia on the ash rachises overwintered in common garden (fructification assay).** (A) Relation with leaf necrosis observed in the same plot two months before (pvalue = 0.304). (B) Relation with climate (pvalue = 0.006, Table 4).

## Discussion

We showed that carrier trees without crown dieback symptoms exist and play a significant role in ash dieback epidemiology (hereafter referred to as "healthy carriers"). *H. fraxineus* is able to establish on ash leaves and to reproduce on the leaf debris in the litter (rachises) in situations where it induces very limited shoot mortality. This can be on ash trees that show no dieback while their neighbors do show some (Figs 2 and 3A), in specific local environments (low tree cover, Fig 3B) or meso-climates (Fig 6). The climate strongly influences *H. fraxineus* with different parameters being important depending on the life cycle stage (Figs 4 and 5). As a consequence, while the pathogen might be able to establish over most France, it should to induce far less severe dieback in large parts of southern France (Fig 6). This is likely to occur in others areas of southern Europe. Last, we observed a decreasing trend of severity (leaf necrosis and shoot mortality frequency) with the time of disease presence (Fig 4D and 4H).

We were able to predict the severity of dieback in spring by observing leaf symptoms in the previous autumn. Leaf necrosis frequency in September was a good predictor of subsequent shoot mortality (Fig 3). Despite that, leaf necrosis was a very poor predictor of rachises infection at leaf fall and of the ability of *H. fraxineus* to produce apothecia, and therefore inoculum,

**Table 4. Influence of climatic parameters on the ability of *H. fraxineus* to produce apothecia in the fructification assay.**

| Variable | Parameter | Pvalue |
|---|---|---|
| Leaf necrosis frequency | -0.44 | 0.297 |
| Rain in July-August (RA78) | 1.23 | <0.001 |
| Mean maximal daily temperature in July-August (TX78) | -0.29 | 0.353 |
| Rain in October-March (RA10-3 ow) | 0.21 | 0.478 |
| Mean temperature in October-March (TM10-3 ow) | 0.55 | 0.089 |
| Site tree cover (%, TC) | 0.11 | 0.549 |
| Site random factor (sd) | 0.0003 | - |

Note: RA10-3 ow and TM10-3 ow, sum of rain and mean of daily temperatures at the overwintering sites from October to March.

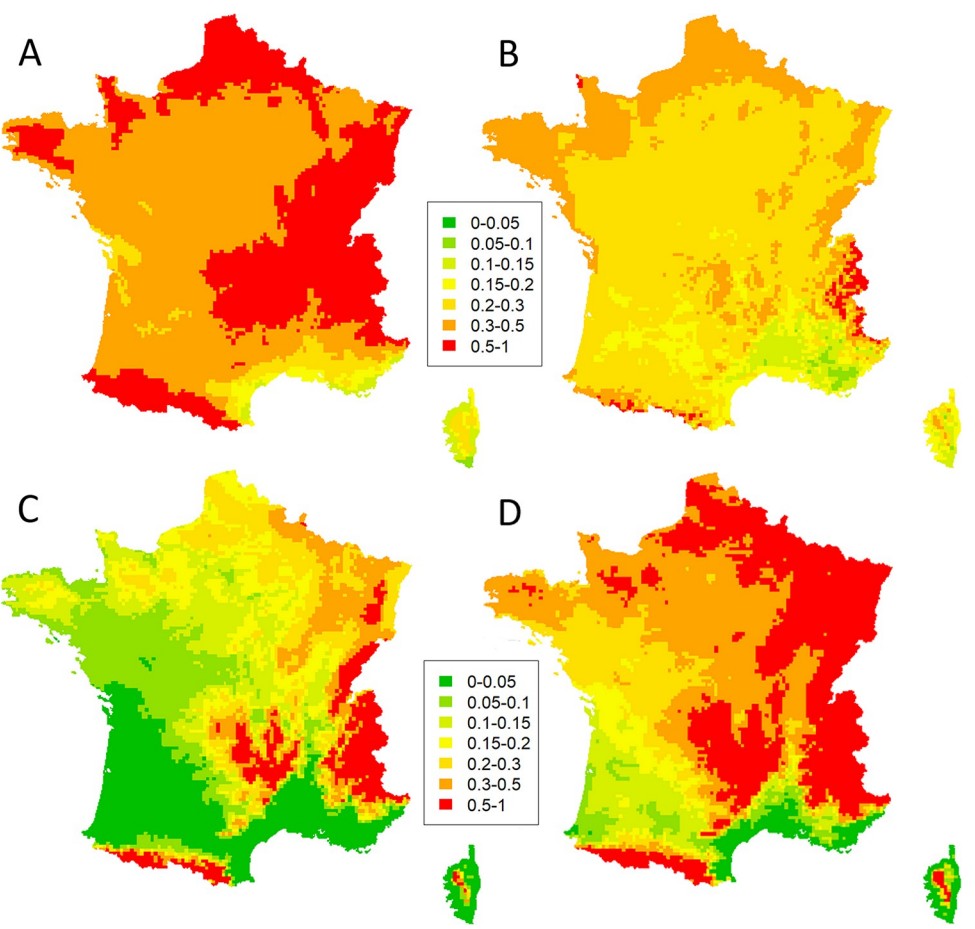

**Fig 6. Map of climate suitability for *H. fraxineus* ability to produce apothecia on the ash rachises or symptoms on leaf and shoot of ash trees.** Safran meteorological data, 2010–20. (A). Proportion of ash rachises collected in fall producing apothecia in laboratory conditions after overwintering in a common garden (fructification assay). (B) Leaf necrosis probability. (C) Shoot mortality probability in open canopy (20% tree cover). (D) Shoot mortality probability in forest conditions (100% tree cover).

on the rachises in spring (Figs 5 and 7). This may be because of the two-month delay between leaf necrosis measurement and rachises colonisation assessment. It has to be pointed out that very little inoculum production is observed in September and October in French conditions [22]. Nevertheless, lesions present in the leaflets might have extended to the rachises. Another likely explanation is that *H. fraxineus* has the ability to asymptomatically infect ash leaves in early summer and to behave as a latent pathogen for an extended period [15]. These authors showed that leaf necrosis occurs only when *H. fraxineus* inoculum reaches a certain level in the leaf which happens at the same time as leaf senescence and of a change in the leaf microbiota composition. It would appear that the high *H. fraxineus* levels in leaves required for leaf necrosis also strongly promote shoot infection from the petiole (Fig 3). By contrast, high *H. fraxineus* levels may not be required for rachis colonisation at leaf fall in senescent tissue. This would explain why high rachis colonisation may be observed in situations with limited previous leaf necrosis (Table 3, Fig 5A).

Because high rachis colonisation may occur while symptom frequency, leaf necrosis or shoot mortality, is low (Table 3), *H. fraxineus* is able to complete its biological cycle in conditions where no significant dieback is observed and some ash trees behave as healthy carriers

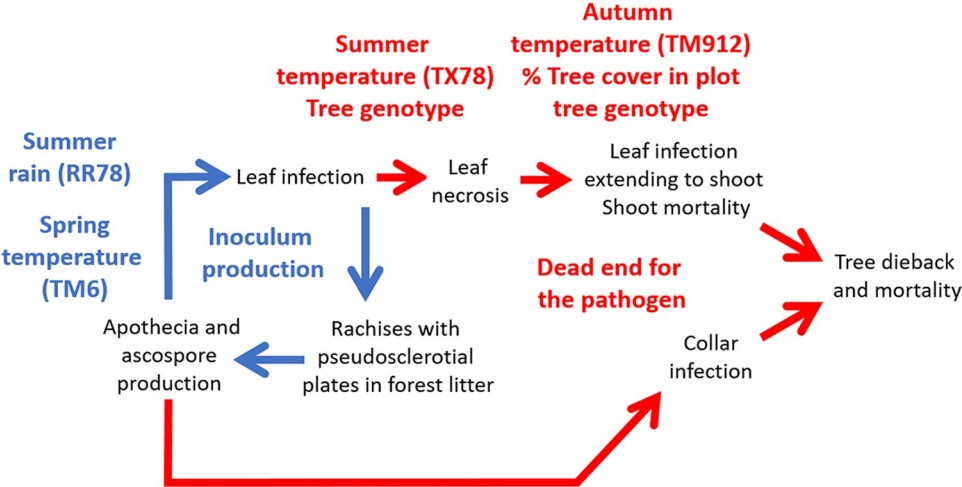

**Fig 7. Hypothesised impact of studied environmental parameters on _H. fraxineus_ life cycle.** Higher summer rainfall and higher spring temperatures are associated with higher rates of leaf necrosis leading to higher shot necrosis especially in high tree cover. Higher summer temperatures are associated with low leaf necrosis frequency at the end of the summer, but not to low colonisation of rachises at leaf-fall during autumn. Low autumn temperatures are associated with high shoot mortality (better transfer from leaves to shoot or lesion development in shoots?). The part of the cycle needed for inoculum production is indicated in blue while red arrows and text refers to symptoms development that does not lead to inoculum production. Cycle adapted from [41].

(Fig 7). This is the case at local level, on specific trees that remain without shoot mortality despite experiencing leaf necrosis and enabling the pathogen to complete its cycle on infected rachises (Fig 2). We did not find a difference in leaf necrosis frequency between saplings showing dieback or not (Table 1). The measure of leaf infection might not have been optimal as we only rated the presence of necrosis on the leaf without any measure of necrosis severity such as number of affected leaflets. Also, we examined only branches within reach to assess the disease severity, which may not be representative of disease severity higher in the crown. Leaf infection might be higher close to soil level as spore load in the air was shown to decrease from 1 to 3 m above soil level [23] and because microclimate is more humid close to the soil and thus more favourable to _H. fraxineus_ infection.

Our experimental design did not allow us to determine whether certain ash trees showed no dieback for genetic reasons or because they were in microsites unfavorable to the disease or harbored foliar microbial competitors preventing _H. fraxineus_ development. Resistance to _H. fraxineus_ has been widely documented in _F. excelsior_ [17,24] and thus remains a very likely explanation for at least part of the observed pattern, although different micro-climatic conditions can be another one (Fig 3A and 3B). It does not appear that some of the studied trees remained without dieback because they avoided inoculum as they showed a frequency of leaf necrosis similar to trees that showed dieback symptoms in the same plots (Figs 2 and 3). In the two stands studied in more detail in 2014, the ash density was high and the saplings remaining without dieback were in close proximity to dying saplings, questioning the possibility that they were experiencing different environmental conditions.

Nevertheless, environmental conditions strongly influenced the ability of _H. fraxineus_ to infect the shoot from the leaf petiole. Indeed, although the frequency of leaf necrosis did not depend on the tree cover, the subsequent frequency of shoot mortality was much lower in hedges with a low tree cover compared to forest plots with a high tree cover (Table 1, Fig 3B). We also showed that the colonisation of rachises at leaf fall and subsequent ability of _H. fraxineus_ to complete its cycle on the colonised rachises does not depend on the tree cover

(Table 3), being similar in forests conditions and in hedges in open landscape. Hence, trees with no dieback symptoms in infected stands behave as heathy carriers, allowing *H. fraxineus* to reproduce on their leaves as efficiently as on the leaves of neighboring trees that experience dieback. They should even be much better inoculum producers than trees affected by dieback as they retain full foliage and thus produce abundant infected rachises. This result is in agreement with the observation of Grosdidier et al. [21] that the density of infected rachises was similar in the litter beneath ash trees in hedges and in forest settings although the trees showed far less dieback symptoms in the latter situation. A possible explanation is that as the tree cover does not affect inoculum production, it does not affect infection frequency. However, at low tree cover, crowns are exposed more frequently to micro-climatic conditions adverse to *H. fraxineus* (hotter and dryer conditions, [21]), preventing the increase of the pathogen present as endophyte in leaves and thus later shoot mortality. This discrepancy between the ability to reproduce on the host and the ability to induce significant damages has been reported for other tree pathogens. For example, *Diplodia sapinea* mainly produce inoculum on pine cones and is widely present on cones in healthy pine stands; it however needs weakening of the trees to induce significant damages [25,26].

At a larger scale, we showed that *H. fraxineus* may be able to fulfil its cycle in regions were climate does not allow significant dieback (Fig 6). The disease has been progressing very regularly in France from 2009 to 2016 with a very noticeable slowdown of the spread in southern France from 2017 to 2019 (Fig 1). The slowdown of the spread occurred in regions with climate unfavorable to ash dieback. In 2020, the Pyrenean area was colonised while no sign of the disease had been reported north in the Garonne valley (data from the Département de la Santé des Forêts). It is possible that the pathogen was introduced via infected planted ashes [27]. However, our results show that the spread without dieback symptoms of *H. fraxineus* in the Garonne valley is also a likely hypothesis. This may have been enhanced by the very hot summers of 2018–20, which were even less favorable to *H. fraxineus*-induced shoot mortality. The delay of 3–4 years observed until the first report of ash dieback in the Pyrenean piedmont is in line with the speed of spread observed in France of about 60 km per year [28].

The decreasing trend of leaf necrosis and shoot mortality frequency with years of disease presence may be very significant but requires confirmation. It may well be a spurious result caused by the very dry and hot climate experienced in northern and eastern France in 2018–20, at the end of the study period. Climatic conditions were taken into account in our model and high summer temperature was one of the most important feature to explain symptom severity. But the parameter we used (average daily maximal temperature) may not have accounted for the entire effect of heat waves observed during this 2018–20 period. Another possible explanation is the progressive selection of more tolerant or resistant ash trees in sites affected by ash dieback for a long time. Indeed, in the site followed for the longest period, fewer than 50% of the saplings selected in 2015 were still alive in 2021. We tried to replace dead saplings by symptomatic saplings to avoid a progressive increase in tolerant/resistant ash frequency in our sample, but this was not always possible.

Early leaf senescence was hypothesized to be a mechanism of tolerance to *H. fraxineus*, the pathogen then being shed in the leaves before getting the chance to infect shoots [17,19]. We were not able to confirm the hypothesis as early defoliation was associated with higher shoot mortality in our data. However, care has to be taken when interpreting this result. Mc Kinney et al. [17] reported that some ash trees show early leaf senescence as a genetically controlled trait, and thus show early defoliation, limiting the transfer of *H. fraxineus* from leaves to shoots. In our experiment, early defoliation was a result of infection, being tightly associated with high leaf necrosis frequency. In these conditions, transfer of the pathogen from leaves to shoots may not be hampered.

Climate appears to be a major driver of symptom induction by *H. fraxineus*. High summer temperatures was expected to limit disease severity, as the pathogen has been shown to have low survival at temperatures above 35˚C [29]. Moreover, the hot summers of the Rhône valley south of Lyon limit ash dieback severity in South-East France [30]. More surprising is the low correlation of spring and summer rainfall with disease severity as it has been associated with high disease impact through increased inoculum production [31,32,33]. Spring and summer rainfall may promote infection of the leaves while *H. fraxineus* remains as an asymptomatic endophyte especially if high summer temperatures prevent the heavy colonisation of ash leaves by the pathogen. This could explain why spring and summer precipitations were not well correlated with frequency of symptoms such as leaf necrosis and shoot mortality (Table 2). By contrast, summer rainfall was strongly associated with both high rachis infection and ability to produce apothecia in the following spring. For example, in 2019, very limited leaf necrosis occurred in the Champenoux area because of high summer temperatures although summer rainfall allowed good colonisation by *H. fraxineus* of the rachises shed during the autumn (Table 3). This might be the likely mechanism by which spring and summer rainfall favour high ash dieback impact. The period in spring with the best fit between climatic conditions and leaf necrosis frequency was in June, although the difference with other tested periods was low. This may appear surprising as the period of ascospores production has often been shown to be in summer, in July to August [23,32,34,35]. However, many of those results correspond to central or northern Europe; in North-East France, the period of ascospores production was earlier, from mid-June to mid-July [22] and ascomata have been observed as early as the first week of May in the Pyrenean sites during this work. Shoot mortality was associated with previous leaf necrosis, but also with subsequent mild autumn temperature. This is the first report of an association of shoot mortality with autumn climatic conditions and the mechanism involved remains to be elucidated.

*H. fraxineus* ability to infect ash leaves and reproduce on the rachises in the forest litter does not appear to limit the pathogen ability to establish throughout France. By contrast, the low ability to induce shoot mortality explains the absence of reports of the disease in many areas of southern France. Hence, the maps showing the shoot mortality probability match well with the ash dieback risk predicted by Goberville et al. [36]. The statistical niche model produced by these authors was based on the distribution in Europe of this invasive pathogen still spreading and with a presence still limited to North-Easter France; it nevertheless well captured the potential climatic niche in France. This would confirm that south-eastern France is not favorable to ash dieback, as was suggested by Grosdidier et al. [30].

## Material and methods

### Sampled plots

We selected 20 plots on a temperature gradient in France (Fig 1, S1 Table) in order to have a range of summer temperature as large as possible within the range *of H. fraxineus* presence. The plots were usually studied several years in a row starting from 2013 in the area where ash dieback was first observed in 2009–10 (NE France) to 2020 in an area just recently affected by ash dieback (SW and NW France, in Brittany and Pyrenean Piedmont). The studied plots were either in forest settings or located in hedges. The ash species present was *Fraxinus excelsior* in all plots. A summary of observations of leaf necrosis and shoot mortality on the plots is given in S1 Table. Plots were characterized by their tree cover from aerial photographs (Orthophoto with 50cm pixels from IGN, http://professionnels.ign.fr). The tree cover polygons within a radius of 100 m from each plot, including all tree species, were extracted and the proportion of tree cover within that range (TC) was computed.

## Survey procedure

In all studied plots, 20 ash trees were marked. In each plot, we selected suitable saplings that were contiguous, marking all those with two shoots available for rating. We usually selected saplings 2–4 m high, although occasionally, larger trees with low-lying branches were selected. Preliminary analysis showed that the tree height did not influence leaf necrosis and shoot mortality probability. Saplings of different health conditions were usually present in the plots, with a proportion of saplings without dieback of shoots varying from 20 to 75% mostly depending on the length of ash dieback presence in the area. This proportion was adequate for the hypothesis we wanted to test. Dieback only refers to the mortality of perennial organs, here the shoots; symptoms occurring on the leaves are not included. The saplings were rated annually for their crown status according to five dieback scores: 0 for complete absence of dead twigs, 1 for less than 10% of dead twigs, 2 for 10 to 50% of dead twigs, 3 for more than 50% of dead twigs and 4 for dead saplings [21]. Two annual shoots per individual were marked for rating of leaf and shoot infection. We selected the terminal shoot of the 2 lower branches that could be reached by an observer from the ground. As most saplings were 2–4 m high, this should not introduce any bias. Each sapling was rated for several years in row (1 to 6 years depending on the site and on the year of first ash dieback report, S1 Table). Dead twigs were replaced by the nearest shoot available from the lower part of the sapling. Dead saplings were replaced by the closest appropriate sapling, taking care to select saplings with ash dieback symptoms to avoid ending up with only individuals potentially tolerant or resistant to *H. fraxineus*. On average, about 18% of the saplings were replaced over the study period, mostly saplings showing dieback as they suffered high mortality.

The necrosis induced by *H. fraxineus* on leaves was rated in September. The total number of leaves present on each marked annual shoot, the number of leaves with lesions (not taking into account the number of affected leaflets) and the number of leaf scars indicating defoliation were determined (altogether 4 to 27 leaves depending on the shoot). Lesions induced by *H. fraxineus* can be recognized as brown lesions starting at leaf margin and extending preferentially on leaf veins. Leaves presenting brown lesions only on the rachis were also rated as infected. Two leaves rated as presenting *H. fraxineus*-induced lesions were sampled for further analysis in the laboratory in 2018–20. Sampled leaves were stored individually in paper bags and kept at ambient temperature. In the laboratory, they were surface sterilized (5 min in 4% NaOCl and 1 min in 90% ethanol) and analysed by qPCR for the presence of *H. fraxineus* as described by Ioos et al. [37]. In the following year's vegetation season, the status of marked shoots was assessed as living or dead. Whenever the shoot had wilted because of girdling occurring below the annual shoot, we considered that it was not possible to determine whether infection had transferred from the annual shoot leaves to the stem and the data was counted as missing.

In October-November, 20 rachises that had just fallen to the ground were sampled by site in order to assess the ability of *H. fraxineus* to produced apothecia on them (called hereafter fructification assay). The rachises were sampled at the base of the marked saplings (one rachis per sapling). This was done in all sites of Champenoux (2016, 2017, 2019, 2020), Fréchencourt (2016, 2017), Lupé (2016), Roche-sur-Grane (2017) and in the 4 Pyrenean sites (Ol1, Ol2, Fe, Be, 2020). The rachises were placed in mesh bags and overwintered in a common garden in a forest site in Champenoux (6.34251, 48.75297, EPSG 4326) except for the rachises of the plots close to Pau that were overwintered close to the site of Sarrance (-0.59692, 43.00044, EPSG 4326). This was done to minimize travel; climatic conditions during the vegetation season conditions determine inoculum production and leaf infection and we supposed that it would be the main factors controlling colonisation of the rachises. In the following spring, in April or

May depending on the year, the rachises were retrieved from the mesh bags and were placed for 6–8 weeks at 18–20°C in humid chambers in large plastics boxes in contact with moistened filter paper to assess their ability to produce apothecia. The moistened filter paper was kept wet throughout the incubation and was changed weekly. Whenever a rachis produced *H. fraxineus* apothecia, the rachis part covered with pseudosclerotial plate was removed and stored in paper bags for further analysis. The part without pseudosclerotial plate was left in the humid chamber. *H. fraxineus* apothecia can be recognized as 2–4 mm white apothecia with a dark base of the pedicels and smooth margin of the fruiting disk [38]. The monitoring was stopped whenever no new production of apothecia was observed in a week (after 6–8 weeks incubation). Remaining rachises were then placed in separate bags. The amount of rachises parts producing / not producing apothecia was determined by measuring the rachis length in 2016 and 2017 and by measuring their weight after drying 3d at 50°C in 2019 and 2020. The proportion of rachises producing apothecia was then computed.

In 2019 and 2020 in sites in Champenoux and in 2020 in Pyrenean sites, 10 additional rachises were collected per site at the same time at leaf fall to determine the frequency of *H. fraxineus* by qPCR. Individual rachises were kept separate. They were dried 3d at 50°C, cut in small 1–2 mm sections and grinded with a mortar. A subsample of 50 mg per rachis was then used for DNA extraction with DNAeasy Plant Mini kit (Qiagen). AP1 lysis buffer (800 μl), 4 μl RNase, two 3-mm and twenty 2-mm glass beads were added to the tube and the sample was grounded twice using a beadbeater (Tissue Lyser, Qiagen) at 30 Hz for 30 s. The DNA extraction was then conducted according to the manufacturer's recommendations. Total DNA was eluted in 150 μl AE buffer. The samples were then analysed by qPCR for the presence of *H. fraxineus* as described in Ioos et al. [37].

## Ability of *H. fraxineus* to complete its cycle on ash saplings with different dieback severity

The ability of *H. fraxineus* to complete its cycle on ash saplings with no dieback symptoms was studied in two of the plots, Gremecey and Seichamps. Ash saplings about 2–3 m high were selected during summer 2014 in order to have 10–13 saplings of two types in each stand: i. saplings with either no or only very few scattered small dead shoots (no dieback, crown status of 0 or 1), ii. saplings strongly affected, with top shoots killed by *H. fraxineus* (crown status of at least 3). The two stands were densely stocked naturally regenerated ash stands heavily and uniformly infected by *H. fraxineus* for at least 4 years. The saplings without dieback were in close vicinity to individuals with severe dieback, with usually several severely affected saplings within less than 1 m from the healthy-looking sapling.

On each sapling, an annual shoot with 7–28 leaves was selected and marked at the end of July. The selected shoot infection by *H. fraxineus* was rated at the beginning of September 2014 with a record of the number of leaves with necrosis and of the total number of leaves. To measure the ability of *H. fraxineus* to complete its cycle on rachises, we placed a net around the entire crown of each sapling in mid-September to collect all the leaves falling during the autumn. The nets containing the leaves were retrieved at the end of October and placed on the ground outdoors in a nursery close to both plots to overwinter. Unfortunately, logging occurred in the Seichamps plot during autumn and a heavy machinery destroyed many studied saplings. Only four saplings without dieback and one with severe dieback were available in that plot for the net harvest. In April 2015, the rachises were retrieved from the 27 nets (11+4 without dieback and 11+1 severely affected by *H. fraxineus*) and were placed in large plastic boxes in contact with moistened filter paper. The boxes were then closed and placed in the laboratory at room temperature. After a 5-week incubation, the rachises were examined for

presence of *H. fraxineus* apothecia. The total length of each rachis was determined as well as the length between the two most distant apothecia of the rachis (hereafter called L.apo). As the part of rachises colonised by *H. fraxineus* was usually covered with many primordia of apothecium in the humid chamber, this is equivalent to the length of rachis colonised by *H. fraxineus*. The percentage of rachis colonisation on a sapling was determined as *Σ L.apo / Σ total rachis length*. The saplings were again assessed in summer 2015 to determine the global infection status and the presence of shoot infection on the marked shoot assessed in 2014 for leaf infection.

The likelihood of leaf necrosis in September 2014 and shoot infection in summer were assessed by logistic regression with the plot, the dieback status of the saplings and their interaction as independent variables. We used a quasibinomial distribution to analyse leaf necrosis likelihood because data showed a dispersion higher than expected for a binomial distribution (over dispersion). The percent rachis colonisation by *H. fraxineus* in spring 2015 was analysed by beta regression using the betareg package. The stand*dieback status interaction was assessed for leaf necrosis but not for shoot infection in 2015 and ability of rachises to produce apothecia as very few saplings then remained in Seichamps.

## Data analysis

During survey, we noticed that some of the rated shoots were very vigorous with many rated leaves and that they were seldom infected by *H. fraxineus* whatever the general situation in the stand. These very vigorous shoots were usually sprouts from the collar of severely affected saplings. Preliminary analysis showed that shoots with more than 20 leaves had on average 5.9% necrotic leaves and no shoot mortality, compared to average values of 16.0% of necrotic leaves and 21.6% shoot mortality for the total dataset. Shoots with more than 20 leaves were thus removed from the data set prior to analysis (represent only 1.5% of the data). Two different models were developed. Bayesian models were fitted with Jags 4.3.0 emulated with R4.1.0 (R2jags library).

## Relationship between leaf necrosis and shoot mortality

The first model aimed at exploring the relationship between leaf necrosis and shoot mortality at the individual shoot level and how this may be controlled by tree health and site tree cover. Data were thus kept at the shoot level for this model. A hierarchical Bayesian model was developed to assess symptoms on shoot n of sapling i in year t:

$$logit(pl[n,i,t]) = \alpha_1 + \alpha_2 \times DS[i] + \alpha_3 \times TC[i] + Tree.l\ [i] + Env.l\ [i,t]$$

$$logit(ps[n,i,t]) = \beta_1 + \beta_2 \times DS[i] + \beta_3 \times TC[i] + \beta_4 \times DEF[n,i,t] + \beta_5 \times pl[n,i,t] + Tree.s\ [i] + Env.s\ [i,t]$$

with pl[n,i,t] and ps[n,i,t] the respective probability of leaf necrosis and shoot mortality, DS[i] the ash dieback status (either no dieback, when crown rating was of 0 or 1 or with dieback, when crown status was of 2 or 3), TC[i] the percent tree cover within a radius of 100 m, DEF[n,i,t] the proportion of missing leaves at the autumn rating, Tree.l [i] and Tree.s [i] individual tree random factors and Env.l [i,t] and Env.s [i,t], plot[i]*year[t] random factors. The Env.l [i] and Env.s [i] random factors were taken as proxies for the environment, combining meso/micro climate and plot conditions (host density, Infection history). The ash dieback status DS[i] was computed as the worst status observed on the sapling during the period of observation (from 2 to 7 years depending on the sapling). The defoliation level (DEF[n,i,t]) was assumed to influence the shoot mortality probability because Mc Kinney et al. [17] suggested that early defoliation may prevent the transfer of *H. fraxineus* from leaves to shoots and thus represent a tolerance mechanism.

Both pl[n,i,t] and ps[n,i,t] were assumed to follow binomial distribution. Non-informative priors were assigned to the model parameters according to a normal distribution N(0, 1e+05) or a uniform distribution U(0,100) for the standard deviation of the random factors Tree and Env. The model was fitted with a burn-in of 3000, for 50000 iterations with a thinning of 5. Three parallel chains with different initial parameter values were run and convergence was checked by a Gelman-Rubin test ($R_{hat}$ were close to 1). The model was checked by looking at the plot of deviance residuals versus predicted values.

## Relationship between leaf necrosis or shoot mortality and climate

The second developed hierarchical Bayesian model aimed at exploring the relationship between leaf necrosis or shoot mortality with climate. For that model, data were pooled at the plot * year level. Like in the previous model, shoot mortality was assumed to derive from former leaf infection and the proxy for leaf infection was taken as the leaf necrosis probability. Length of ash dieback presence in the area was included in the model to explain the probability of leaf necrosis. The year of first report of ash dieback by the health survey system is available by 16 x 16 km quadrat over France from data of the health survey system (Département de la Santé des Forêts, Fig 1). The number of years since the report of ash dieback in the local quadrat was taken as a proxy for the length of the disease presence. It was considered an adequate proxy as local spread of *H. fraxineus* at the scale of a village was shown to be very fast [21]. Leaf necrosis probability was assumed to depend on spring and summer climatic conditions (total rainfall and average temperatures). It is known that *H. fraxineus* may not survive well during hot summers [29]. In particular, this pathogen is known to stop its growth at temperatures above 28°C and to have limited survival when temperature exceeds 36°C. To account for this, we used TX78, the average daily maximal temperature in July-August (°C). Shoot mortality was assumed to depend on autumn climatic conditions (total rainfall and average temperatures). The meteorological data used were Safran data from Météo-France. Safran data are computed on a 8 x 8 km grid over France [39]. The Safran point closest to each site was selected and daily air temperature and rainfall data were retrieved for the studied years. All predictors were normalised to obtain zero mean and one standard deviation. However, the odds ratio reported in the text are in the non-normalized scale.

The following hierarchical Bayesian model was fitted [40]:

$$\text{logit}(pl[i,t]) = \alpha_1 + \alpha_2 \times RA6[i,t] + \alpha_3 \times TM6[i,t] + \alpha_4 \times RA78[i,t] + \alpha_5 \times TX78[i,t] + \alpha_6 \times LDP[i,t] + Site.l\ [i]$$

$$\text{logit}(ps[i,t]) = \beta_1 + \beta_2 \times RA912[i,t] + \beta_3 \times TM912[i,t] + \beta_4 \times TC[i] + \beta5 \times pl[i,t] + Site.s\ [i]$$

with pl[i,t] and ps[i,t] the respective probability of leaf necrosis and shoot mortality in plot i and year t, RA6**[i,t] and** TM6**[i,t]** the sum of rain and mean daily air temperature in June, RA78**[i,t] and** TX78[i,t] the sum of rain and average daily air maximal temperature in July-August (°C), **LDP**[i,t] the number of years of ash dieback presence in the area, RA912**[i,t]** and TM912[i,t], the sum of rain and mean daily air temperature in September, October, November and December and TC[i] the percent tree cover within a radius of 100m. Site.l [i] and Site.s [i] are site random factors, taken as a proxy for the site environment (micro climate, site conditions such as host density or infection history).

While ps[i,t] was assumed to follow a binomial distribution, pl[i,t] was assumed to follow a beta distribution as the number of observed leaves per site was very high (from 200 to 400). Non-informative priors were assigned to the model parameters according to a normal distribution N(0, 1e+05) or a uniform distribution U(0,100) for the standard deviation of the random factors. The model was fitted with a burn-in of 10000, for 80000 iterations with a

thinning of 10. Three parallel chains with different initial parameter values were run and convergence was checked by a Gelman-Rubin test ($R_{hat}$ were close to 1).

The suitable periods for spring and autumn were selected in a preliminary analysis. The explored periods targeted either the period of ascospore production in spring or of the development of the shoot necrosis during the autumn. Several periods were tested: from 15 May to 15 June, for the 1 to 30 June or from the 15 June to 15 July for spring and September to October or September to December for autumn. The model showing the best deviance information criteria (DIC) was selected.

The final model was checked by looking at the plot of deviance residuals versus predicted values and by a cross validation procedure. Data were randomly separated in 5 equal groups and the model was fitted using the data from 4 of the groups; predicted leaf necrosis frequency and shoot mortality were computed for the fifth group not used for the fit. The procedure was repeated five times and the observed values obtained by the procedure were plotted against the predicted values.

### Relationship between climatic parameters and ability of *H. fraxineus* to produce apothecia in the fructification assay

The proportion of rachises producing apothecia in the fructification assay was analysed by beta regression using the glmmTMB library. The site was included as random factor. Fixed factors were the proportion of leaves with necrosis at the September rating, climatic parameters in the studied sites during the summer (total rainfall and mean of the daily maximal temperature in July and August) and climatic parameters during the overwintering period (average temperature and total rainfall at the overwintering locations from the beginning of October to the end of March).

## Supporting information

**S1 Table. Studied sites.**
(DOCX)

**S1 Fig. Cross-validation for the model at the plot level relating climatic and site parameters to leak necrosis and shoot mortality probability.** A. Leaf necrosis, B. shoot mortality.
(DOCX)

**S2 Fig.** Quantile 0.05 (Q5) and 0.95 (Q95) for the estimated probability of leaf necrosis (A, B) and shoot mortality (C, D). The map were computed using the parameter 0.05 and 0.95 quantiles obtained from the Bayesian procedure fit. The shoot mortality is computed for a forest situation (tree cover of 100%).
(DOCX)

## Acknowledgments

We wish to thank Olivier Caël and Anaïs Gillet for technical assistance in handling the analysis of rachises ability to produce apothecia and level of infection by *H. fraxineus*. The establishment and rating of the plots in the Charente area was done by Dominique Piou while those of plots from Brittany was done by Laurence Roche from the Département de la Santé des Forêts. We also thank Mireia Gomez-Gallego for her advices in the analysis and for reviewing the manuscript.

## Author Contributions

**Conceptualization:** Benoit Marçais, Claude Husson.

**Data curation:** Arnaud Giraudel.

**Formal analysis:** Benoit Marçais, Arnaud Giraudel.

**Funding acquisition:** Benoit Marçais.

**Investigation:** Benoit Marçais, Arnaud Giraudel, Claude Husson.

**Methodology:** Benoit Marçais.

**Writing – original draft:** Benoit Marçais.

**Writing – review & editing:** Arnaud Giraudel, Claude Husson.

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
