## [Decision Letter · Decision Letter 0]

8 Jul 2022

Dear Dr Marçais,

Thank you very much for submitting your manuscript "Ability of the Ash dieback pathogen to reproduce and to induce damage on its host are controlled by different environmental parameters" for consideration at PLOS Pathogens. As with all papers reviewed by the journal, your manuscript was reviewed by members of the editorial board and by several independent reviewers. In light of the reviews (below this email), we would like to invite the resubmission of a significantly-revised version that takes into account the reviewers' comments.

Thank-you for submitting this paper to PLOS Pathogens. I've received reports from two specialist reviewers who agree that your paper is novel and interesting, and potentially has significant practical value. Both of them have quite a long list of points which should be checked or which need further explanation or clarification; please see their copious notes as comments on your PDF. When you submit a revised version of this paper, please include an itemised response to their comments.

You will see that neither reviewer feels fully capable of assessing your Bayesian modelling. I believe I have more experience of Bayesian methods than either reviewer although I'm not a Bayesian specialist. Your model seems fine: appropriate to your data and your hypotheses. However, please consider giving some additional, brief explanation of key points; this is mentioned by Reviewer 1 in particular. For example, explain why you need to use thinning: is that because of autocorrelation within the Markov chain caused by the model being rather complex?

Reviewer 1 mentions the distinction between resistance and tolerance. I think you're aware that this has become confused in the ash dieback literature because a few authors have used the word 'tolerance' when they're actually describing quantitative resistance. Your use of 'tolerance' seems to be in accord with standard usage in plant pathology for nearly 60 years, i.e. a reduction in the amount of disease caused by a certain quantity of the parasite, whereas resistance is a reduction in the amount of the parasite and thus the amount of disease. This means it's correct to describe "healthy carriers" as tolerant. Please would you consider Reviewer 1's concerns about your use of 'tolerance' and 'resistance' and check that your usage is clear and correct. Please conform to the standard usage in plant pathology, and and make sure that you don't say tolerance when you really mean quantitative or partial resistance.

I share both reviewers' concerns about the clarity of some of the text. For example, a sentence which I still don't understand after several readings is near the start of the Results: "The percent of leaves with necrosis in early September 2014 was similar on asymptomatic

and severely affected saplings". How can asymptomatic plants have necrosis?! Perhaps this is because, as you say in the Materials & Methods, asymptomatic saplings may have "either no or only very few scattered small dead shoots". In this case, I suggest you need an alternative word to 'asymptomatic', which suggests there are no symptoms at all. It would be good if you could check the clarity and rigour of your text throughout.

I very much look forward to reading the revised version of this paper.

We cannot make any decision about publication until we have seen the revised manuscript and your response to the reviewers' comments. Your revised manuscript is also likely to be sent to reviewers for further evaluation.

Yours sincerely,

James K.M. Brown

Guest Editor

PLOS Pathogens

Bart Thomma

Section Editor

PLOS Pathogens

Kasturi Haldar

Editor-in-Chief

PLOS Pathogens

orcid.org/0000-0001-5065-158X

Michael Malim

Editor-in-Chief

PLOS Pathogens

orcid.org/0000-0002-7699-2064

Thank-you for submitting this paper to PLOS Pathogens. I've received reports from two specialist reviewers who agree that your paper is novel and interesting, and potentially has significant practical value. Both of them have quite a long list of points which should be checked or which need further explanation or clarification; please see their copious notes as comments on your PDF. When you submit a revised version of this paper, please include an itemised response to their comments.

You will see that neither reviewer feels fully capable of assessing your Bayesian modelling. I believe I have more experience of Bayesian methods than either reviewer although I'm not a Bayesian specialist. Your model seems fine: appropriate to your data and your hypotheses. However, please consider giving some additional, brief explanation of key points; this is mentioned by Reviewer 1 in particular. For example, explain why you need to use thinning: is that because of autocorrelation within the Markov chain caused by the model being rather complex?

Reviewer 1 mentions the distinction between resistance and tolerance. I think you're aware that this has become confused in the ash dieback literature because a few authors have used the word 'tolerance' when they're actually describing quantitative resistance. Your use of 'tolerance' seems to be in accord with standard usage in plant pathology for nearly 60 years, i.e. a reduction in the amount of disease caused by a certain quantity of the parasite, whereas resistance is a reduction in the amount of the parasite and thus the amount of disease. This means it's correct to describe "healthy carriers" as tolerant. Please would you consider Reviewer 1's concerns about your use of 'tolerance' and 'resistance' and check that your usage is clear and correct. Please conform to the standard usage in plant pathology, and and make sure that you don't say tolerance when you really mean quantitative or partial resistance.

I share both reviewers' concerns about the clarity of some of the text. For example, a sentence which I still don't understand after several readings is near the start of the Results: "The percent of leaves with necrosis in early September 2014 was similar on asymptomatic

and severely affected saplings". How can asymptomatic plants have necrosis?! Perhaps this is because, as you say in the Materials & Methods, asymptomatic saplings may have "either no or only very few scattered small dead shoots". In this case, I suggest you need an alternative word to 'asymptomatic', which suggests there are no symptoms at all. It would be good if you could check the clarity and rigour of your text throughout.

I very much look forward to reading the revised version of this paper.

Reviewer's Responses to Questions

**Part I - Summary**

Reviewer #1: This is an interesting paper that demonstrates the clear lifecycle boundaries of biotrophic infection and then saprotrophic reproduction and the environmental parameters which influence these two events. The model produced shows that although ash rachises can produce apothecia over all of France, shoot mortality is limited by climate, particularly temperature. This will limit the disease spread, which may have implications for management practices. Their conclusion suggests that asymptomatic but infected ash in France contribute to the ongoing life cycle of H. fraxineus.

The authors have discussed the concept of a healthy carrier ie. an infected ash that can allow reproduction by doesn't show disease symptoms. The paper also discusses both tolerance and resistance, a subject which has recently become quite muddled in the literature, particularly in relation to ash dieback. To suggest tolerance would mean that they is no fitness cost to ash of being infected with Hf. This has not been studied here; costs could be reduced growth rates, or less seed production for example. From the description given , the authors discuss resistance. The manuscript would benefit from using term resistance. The term 'healthy carrier' would still apply.

The manuscript would benefit from justifications and clarifications that I have noted throughout the attached marked up review, particularly with reference to the experimental design and analysis. The methods section is unclear in several places (noted on the attachment). The authors need to justify leaving out samples/observations and for declaring data missing.

Reviewer #2: The study was well designed and executed too look into the effect of climate and on ADB lifecycle.

Using measurements form a large number of sites across many years enabled the authors to confidently predict the effect of rain fall and temperature on leaf necrosis, defoliation, shoot dieback and reproduction which are very significant contribution to the field. I am not a modelling expert but some potentially interesting interactions between environment and the host class (healthy / declining) was not analysed and could be added.

Another significant contribution of the study is correlations between the different symptoms and ADB reproduction efficiency, which clearly showed the existence of hosts with high inoculum production potential and low or no shoot damage. I have my reservations on the methodology used for the assessment of fructification and have made comments in the text (see attached PDF) for the authors to elaborate on the methodology and its impact on the results/outcomes.

The novelty of the manuscript is considerable but the implications of the climate conditions on the revival of rash across Europe could be discussed. E.g. by prediction of ADB niches in the future using the current climate change models together with overlapping the density of ash in France with disease risks.

The manuscript is written well overall. It only needs minor modifications and additions to improve figures/ tables, add some references and clarify a few points of methodology/results and conclusions.

**Part II – Major Issues: Key Experiments Required for Acceptance**

Reviewer #1: My main concerns are about removing data (such as vigourous growing shoots and using data missing for dead shoots rather than including them in the analysis. It may be that clarifications in the text can resolve these issues if explained properly from a biological reason. It may be that the analysis needs to include these samples.

It is not clear if the interaction between rainfall and temperature has been analysed-I suspect that hot summers are linked to dry summers (or at least les rain days). As presented, the two factors are hard to separate. I would like to see clear evidence that they are not dependent on each other.

Reviewer #2: (No Response)

**Part III – Minor Issues: Editorial and Data Presentation Modifications**

Reviewer #1: The manuscript would benefit from many clarifications which are detailed in the attachment. As noted above, the use of the terms tolerance and resistance are confused, and the manuscript should be reworded to use only resistance, except where tolerance can be justified. I see nowhere in this manuscript where tolerance is proven. The term healthy carrier is useful as it suggest a tree which has little or no disease symptoms but can still produce inoculum on the fallen rachises.

The authors need to provide clarifications on the statistical methods (including what software was used). Many of the results are presented in terms of the statistics, the authors should discuss what these mean in a biological sense. For example : "Both leaf necrosis and shoot mortality were strongly affected by the tree random factor with a similar range of standard deviation" should be explained in non statistical terms. This will make it much easier for a reader without statistical knowledge to understand the manuscript.

As noted in the attachment, some of the graph axis titles are so small they are unreadable even at 150%. These need to be bigger.

The grammar in the paper needs to be improved throughout.

Reviewer #2: All comments are listed directly in the PDF file attached.

PLOS authors have the option to publish the peer review history of their article (what does this mean?). If published, this will include your full peer review and any attached files.

Reviewer #1: No

Reviewer #2: **Yes: **Dr Matevz Papp-Rupar
---

## [Decision Letter · Decision Letter 1]

25 Nov 2022

Dear Dr Marçais,

Thank you very much for submitting your manuscript "Ability of the Ash dieback pathogen to reproduce and to induce damage on its host are controlled by different environmental parameters" for consideration at PLOS Pathogens. As with all papers reviewed by the journal, your manuscript was reviewed by members of the editorial board and by several independent reviewers. In light of the reviews (below this email), we would like to invite the resubmission of a significantly-revised version that takes into account the reviewers' comments.

This is an interesting paper about contrasting effects of environmental variation on pathogen development and disease etiology. The literature on tolerance (the ability of a plant to suppress disease without necessarily suppressing the pathogen itself) is limited and this is one of the most comprehensive attempts to analyse the effects of a range of environmental variables on a pathogen and the disease it causes. It is also important in understanding the difference between tolerance and resistance in a disease with a complex life cycle, and the conclusions have value for managing a serious invasive disease of an important hardwood tree.

That said, it is not an easy read, which is no doubt partly because of the complex nature of the study, involving a wide range of environmental variables, many diverse study sites with differing histories of the disease, and multiple aspects of pathogen biology and disease symptoms. These points are perhaps inevitable in research on a disease with a poorly-understood life history in the natural environment. However, there are quite a lot of places where the text is less clear than it should be, which makes the paper especially challenging, and I agree with Reviewer 1 that the presentation of some of the results and conclusions could be improved significantly. I have recommended numerous clarifications and revisions, especially but not only in the Results and Discussion.

I have commented in detail because of the contrast between the two reviewers' reports on the second version of the paper. In particular, I wanted to check if I agreed with Reviewer 1's comment about a lack of clarity (I do).

Abstract: This is good

Line 14: I suggest, "genetic resistance or tolerance to the disease".

Author summary

Having read your paper thoroughly, I think the last sentence of the Author Summary is one of the most speculative points in the paper and could be omitted. It's potentially misleading for a non-specialist reader because in the paragraph starting on line 511, you acknowledge that this could be an artefact because of a recent run of dry summers in France. If you wish to use the space to mention another important conclusion instead, please do.

Short title: Perhaps, "Environmental influences on ash dieback"? You consider environmental variables other than climate.

The Introduction is very good - an excellent summary of the problem and the related literature.

Line 91: "very limited inoculum production occurs on diseased shoots". This is slightly confusing because "shoots" includes the leaves on those shoots. Maybe, "occurs on the bark of diseased shoots" - is that what you mean?

Line 97: I suggest "fallen leaves" to avoid giving the impression that the pathogens undergoes (sexual) reproduction on green leaves.

Line 100: I suggest something like, "could prevent transfer of the pathogen from infected leaves to shoots but still allow sexual reproduction on fallen rachises". Also see lines 255-258.

Lines 109-114: I feel the first sentence of this paragraph suggests that the emphasis of the paper is on discovering if ash trees are tolerant, while "studied environmental influence" is imprecise. Your data are indeed relevant to tolerance but this is a consequence of what you have discovered about environmental effects on different stages of the disease. I suggest rephrasing line 110 to, "we tested if different environmental variables favour important stages..." I also suggest deleting the last sentence of the paragraph (the key steps of the H.f. life cycle have already been characterised) and replacing it by something like, "We propose that healthy carriers, particularly tolerant genotypes of ash, could be favoured in locations where the environment promotes infection and reproduction of H.f. but not symptom development".

Materials and methods

This is mostly fairly clear but some points could be clarified or made more precise.

Line 120: I suggest, "where ash dieback was first observed in 2009-10" (given that the disease may have been present for many years previously; note Wylder et al. 2018).

Line 126: Does "tree cover" mean all trees or only ash? I believe you mean all species but please make this clear.

Line 131: Was there any difference between saplings and "larger trees with low-lying branches"? Could it have influenced variation between plots?

Line 133: When you say, "without dieback symptoms", do you mean there were no dead twigs or shoots and you discounted lesions on leaves? (Noting that you go on to describe your rating system for saplings in terms of the proportion of dead wood.) If so, I suggest, "without dieback of shoots or twigs".

Important point: Lines 167-8: "we were mainly interested in the climatic conditions during rachis colonisation phase in June –October": does this mean you assumed that environmental conditions were not important during the overwintering stage (do we know what happens at that time - pseudosclerotia maturing, perhaps?) or in spring, when ascomata are developing? Similarly, lines 168-170: You used a standard set of conditions to produce apothecia in April and May, but surely local conditions during this time could be important in the life cycle of H.f.? Your decision to focus on conditions from June to October needs more justification than simply "we were interested".

The point about rating samplings over 1-6 years and the important point about replacing saplings (lines 154-158) is part of the study design. I therefore suggest moving those three sentences to the earlier paragraph where you describe the selection of saplings, e.g. to line 135. I agree with Reviewer 1 that it is not clear if you rated individual saplings over 1-6 years (I think that's what you did but it isn't totally obvious) or if you used a different selection of saplings from each site in each year. This is easily fixed by changing "Saplings were rated" to "Each sapling was rated" on line 154. I also agree with the reviewer that your description of how replacement saplings were chosen is not as clear as it should be. On line 157, when you say "Dead saplings were replaced", were they replaced by the closest appropriate sapling, or by another sapling from somewhere within the same plot, or what? And on line 158, I suggest, "tolerant or resistant to H.frax". You could also say that only 1.5% of saplings in total were replaced over the study period (as I understand from your response to the reviewers of the original version), which indicates that this doesn't affect the majority of the data.

Line 295 (and elsewhere): Did you use a procedure to avoid for Type I errors in multiple significance tests?

Results

I appreciate that you have taken the approach of reporting the results concisely and factually, with little interpretation, which is good practice. However, there are many points where the text is unclear or ambiguous. Please consider the following suggestions for improvement.

Line 337: "very few leaf necrosis": To be clear, did you score the existence or absence of necrosis on a leaf as a categorical variable with two states? Or did you score the number / proportion of leaflets with necrosis within a leaf, or the area of necrosis? If you just scored presence / absence, a point for the Discussion is whether or not this is an accurate measure of disease development on an ash sapling.

Lines 339-340: This sentence doesn't make sense. Do you mean that the frequency of the presence of H.f. in leaves is similar to the visual estimate of the presence of leaf necrosis when necrosis is very frequent, but higher when necrosis frequency is low?

Lines 342-343: I don't understand how the percentage of leaves with necrosis can be similar on a plant with a dieback score of 0 or 1 ("complete absence of dead twigs" or "less than 10% dead twigs") and on one with a dieback score of 3 ("more than 50% of dead twigs"). Are you including the dead twigs in the assessment of % leaves with necrosis? If not (I can only assume you aren't), do you mean that the percent of leaves which are partly necrotic but still partly green is the same on trees with no shoot dieback or extensive shoot dieback? Or was the frequency of necrosis so high that almost all leaves had some necrosis, even on plants with little shoot dieback? At any rate, some major clarification or explanation of this point is required. Referring to my previous point, if the frequency of leaves with necrosis is high even on plants with a low dieback score, this indicates that the necrosis frequency has little power to discriminate healthier and sicker trees, and that a more quantitative score such as the frequency of necrotic leaflets or the area of necrosis would be more precise.

Line 346: "the plot * dieback status interaction was not significant" - I appreciate that this may be an accurate report of a statistical test but it's one of several places where the biological relevance of the statistical result is unclear. Do you mean that the frequencies of different dieback scores were similar in different plots?

Lines 352-353: "The likelihood of shoot infection was significantly higher in [trees?] with severe dieback than in those with no dieback". As written, this seems trivial but do you mean that the likelihood of infection of new shoots was significantly higher in trees with more severe dieback?

Line 355: "Shoot mortality in spring was strongly related to leaf necrosis likelihood". Unclear: why 'likelihood of necrosis'? Why not just the amount of necrosis at the end of the previous summer? A similar comments applies to the sentences on lines 357-8 and 358-60.

Line 360: "leaf necrosis did not significantly depend on tree dieback status". This looks like the same point as on lines 342-3 but the statistics are different. Clarification needed.

Line 367: "affected by tree cover". In which direction? (I can see it in Fig. 3 but please say in the text.)

Lines 394-5: "shoot mortality was strongly linked to leaf necrosis likelihood". (a) In which direction? (As for line 367.) (b) Why 'necrosis likelihood'? (As for line 355.)

Line 395: "shoot mortality ... increased with increasing tree cover". This looks like the same point as you made on line 367. Clarification needed.

Lines 415-6: "rachises producing apothecia ... not significantly related to ... leaf necrosis". In Table 4, there's a very large (negative) parameter relating apothecia to necrosis which isn't significant (P=0.3) but a much smaller (positive) parameter relating apothecia to rainfall, which *is* significant (P=0.006). Are you able to give a reason for this apparent disparity?

Discussion

Having written the Results section in a concise manner with little or no interpretation, it would greatly help readers to understand and evaluate the conclusions of your research if you referred to your figures and tables throughout the Discussion. The study is so complex with many explanatory variables which interact to affect three outcome variables differently (necrosis, mortality, apothecia), and have somewhat different effects at many locations, that it's very challenging to follow the Discussion without a clear guide as to which piece of data is relevant.

For example, your very first point in the Discussion, "healthy carriers exist": this requires the reader to piece together results from several tables and figures. There is some explanation in the next sentence but it would help readers greatly if each point in that sentence had references to the relevant data.

A specific point is that your paper is of at least as much interest to plant pathologists as to ecologists but, unlike ecologist, few pathologists are familiar with bayesian statistics. Without more exposition of your conclusions, I frankly believe that most readers will find the Discussion very difficult to follow in detail. I'm familiar with bayesian methods but I myself found it pretty tough.

Looking back at my notes on the Discussion, most of them just say "Explain" - I think that gives you a flavour of what's needed!

Specific points in the Discussion:

Line 438: Do you have any information which would allow you to extrapolate to other regions, beyond France?

Line 446: Likewise, just in French conditions? From observation, I think this is true in the UK because I've very rarely seen apothecia even in early September. If there's hard evidence for this outside France, please give a reference.

Line 492: You say that at low tree cover, crowns are exposed more frequently to adverse temperatures. Could there also be more wind or a dryer micro-environment, both of which are unfavourable to H.frax? (Extrapolating Chumanova's conclusions to the level of individual trees.)

Line 522-3 (sentence about microbiota): Interesting but rather speculative - omit?

Lines 543-5: I don't understand this sentence. I understand that infection can be promoted by rain but necrosis can be suppressed by high temperatures. But why does this explain why rainfall isn't correlated with symptoms, given that high summer rainfall usually involves lower temperatures in summer (but higher in winter)? Or doesn't the correlation between rainfall and lower temperature in summer apply so much in France?

Lines 558-60: "H. fraxineus ability to infect ash leaves and reproduce on the rachises in the forest litter does not appears to limit the pathogen ability to establish throughout France. By contrast, the ability to induce shoot mortality is the limiting step." This is another point I don't understand because a key point of this paper is that reproduction of H.frax on fallen rachises is not closely connected with shoot mortality. Why then do you say that shoot mortality is a limiting step in the establishment of the pathogen? Especially because in Figure 7, you say that shoot mortality is a dead end for the pathogen.

Tables 1 and 2: Please say that positive values indicate an association between the higher levels of the symptom and higher values of the environmental variable.

Figure 2C: On the y-axis, do you mean % shoots with *leaf* necrosis in spring 2015?

We cannot make any decision about publication until we have seen the revised manuscript and your response to the reviewers' comments. Your revised manuscript is also likely to be sent to reviewers for further evaluation.

Sincerely,

James K.M. Brown

Guest Editor

PLOS Pathogens

Bart Thomma

Section Editor

PLOS Pathogens

Kasturi Haldar

Editor-in-Chief

PLOS Pathogens

orcid.org/0000-0001-5065-158X

Michael Malim

Editor-in-Chief

PLOS Pathogens

orcid.org/0000-0002-7699-2064

This is an interesting paper about contrasting effects of environmental variation on pathogen development and disease etiology. The literature on tolerance (the ability of a plant to suppress disease without necessarily suppressing the pathogen itself) is limited and this is one of the most comprehensive attempts to analyse the effects of a range of environmental variables on a pathogen and the disease it causes. It is also important in understanding the difference between tolerance and resistance in a disease with a complex life cycle, and the conclusions have value for managing a serious invasive disease of an important hardwood tree.

That said, it is not an easy read, which is no doubt partly because of the complex nature of the study, involving a wide range of environmental variables, many diverse study sites with differing histories of the disease, and multiple aspects of pathogen biology and disease symptoms. These points are perhaps inevitable in research on a disease with a poorly-understood life history in the natural environment. However, there are quite a lot of places where the text is less clear than it should be, which makes the paper especially challenging, and I agree with Reviewer 1 that the presentation of some of the results and conclusions could be improved significantly. I have recommended numerous clarifications and revisions, especially but not only in the Results and Discussion.

I have commented in detail because of the contrast between the two reviewers' reports on the second version of the paper. In particular, I wanted to check if I agreed with Reviewer 1's comment about a lack of clarity (I do).

Abstract: This is good

Line 14: I suggest, "genetic resistance or tolerance to the disease".

Author summary

Having read your paper thoroughly, I think the last sentence of the Author Summary is one of the most speculative points in the paper and could be omitted. It's potentially misleading for a non-specialist reader because in the paragraph starting on line 511, you acknowledge that this could be an artefact because of a recent run of dry summers in France. If you wish to use the space to mention another important conclusion instead, please do.

Short title: Perhaps, "Environmental influences on ash dieback"? You consider environmental variables other than climate.

The Introduction is very good - an excellent summary of the problem and the related literature.

Line 91: "very limited inoculum production occurs on diseased shoots". This is slightly confusing because "shoots" includes the leaves on those shoots. Maybe, "occurs on the bark of diseased shoots" - is that what you mean?

Line 97: I suggest "fallen leaves" to avoid giving the impression that the pathogens undergoes (sexual) reproduction on green leaves.

Line 100: I suggest something like, "could prevent transfer of the pathogen from infected leaves to shoots but still allow sexual reproduction on fallen rachises". Also see lines 255-258.

Lines 109-114: I feel the first sentence of this paragraph suggests that the emphasis of the paper is on discovering if ash trees are tolerant, while "studied environmental influence" is imprecise. Your data are indeed relevant to tolerance but this is a consequence of what you have discovered about environmental effects on different stages of the disease. I suggest rephrasing line 110 to, "we tested if different environmental variables favour important stages..." I also suggest deleting the last sentence of the paragraph (the key steps of the H.f. life cycle have already been characterised) and replacing it by something like, "We propose that healthy carriers, particularly tolerant genotypes of ash, could be favoured in locations where the environment promotes infection and reproduction of H.f. but not symptom development".

Materials and methods

This is mostly fairly clear but some points could be clarified or made more precise.

Line 120: I suggest, "where ash dieback was first observed in 2009-10" (given that the disease may have been present for many years previously; note Wylder et al. 2018).

Line 126: Does "tree cover" mean all trees or only ash? I believe you mean all species but please make this clear.

Line 131: Was there any difference between saplings and "larger trees with low-lying branches"? Could it have influenced variation between plots?

Line 133: When you say, "without dieback symptoms", do you mean there were no dead twigs or shoots and you discounted lesions on leaves? (Noting that you go on to describe your rating system for saplings in terms of the proportion of dead wood.) If so, I suggest, "without dieback of shoots or twigs".

Important point: Lines 167-8: "we were mainly interested in the climatic conditions during rachis colonisation phase in June –October": does this mean you assumed that environmental conditions were not important during the overwintering stage (do we know what happens at that time - pseudosclerotia maturing, perhaps?) or in spring, when ascomata are developing? Similarly, lines 168-170: You used a standard set of conditions to produce apothecia in April and May, but surely local conditions during this time could be important in the life cycle of H.f.? Your decision to focus on conditions from June to October needs more justification than simply "we were interested".

The point about rating samplings over 1-6 years and the important point about replacing saplings (lines 154-158) is part of the study design. I therefore suggest moving those three sentences to the earlier paragraph where you describe the selection of saplings, e.g. to line 135. I agree with Reviewer 1 that it is not clear if you rated individual saplings over 1-6 years (I think that's what you did but it isn't totally obvious) or if you used a different selection of saplings from each site in each year. This is easily fixed by changing "Saplings were rated" to "Each sapling was rated" on line 154. I also agree with the reviewer that your description of how replacement saplings were chosen is not as clear as it should be. On line 157, when you say "Dead saplings were replaced", were they replaced by the closest appropriate sapling, or by another sapling from somewhere within the same plot, or what? And on line 158, I suggest, "tolerant or resistant to H.frax". You could also say that only 1.5% of saplings in total were replaced over the study period (as I understand from your response to the reviewers of the original version), which indicates that this doesn't affect the majority of the data.

Line 295 (and elsewhere): Did you use a procedure to avoid for Type I errors in multiple significance tests?

Results

I appreciate that you have taken the approach of reporting the results concisely and factually, with little interpretation, which is good practice. However, there are many points where the text is unclear or ambiguous. Please consider the following suggestions for improvement.

Line 337: "very few leaf necrosis": To be clear, did you score the existence or absence of necrosis on a leaf as a categorical variable with two states? Or did you score the number / proportion of leaflets with necrosis within a leaf, or the area of necrosis? If you just scored presence / absence, a point for the Discussion is whether or not this is an accurate measure of disease development on an ash sapling.

Lines 339-340: This sentence doesn't make sense. Do you mean that the frequency of the presence of H.f. in leaves is similar to the visual estimate of the presence of leaf necrosis when necrosis is very frequent, but higher when necrosis frequency is low?

Lines 342-343: I don't understand how the percentage of leaves with necrosis can be similar on a plant with a dieback score of 0 or 1 ("complete absence of dead twigs" or "less than 10% dead twigs") and on one with a dieback score of 3 ("more than 50% of dead twigs"). Are you including the dead twigs in the assessment of % leaves with necrosis? If not (I can only assume you aren't), do you mean that the percent of leaves which are partly necrotic but still partly green is the same on trees with no shoot dieback or extensive shoot dieback? Or was the frequency of necrosis so high that almost all leaves had some necrosis, even on plants with little shoot dieback? At any rate, some major clarification or explanation of this point is required. Referring to my previous point, if the frequency of leaves with necrosis is high even on plants with a low dieback score, this indicates that the necrosis frequency has little power to discriminate healthier and sicker trees, and that a more quantitative score such as the frequency of necrotic leaflets or the area of necrosis would be more precise.

Line 346: "the plot * dieback status interaction was not significant" - I appreciate that this may be an accurate report of a statistical test but it's one of several places where the biological relevance of the statistical result is unclear. Do you mean that the frequencies of different dieback scores were similar in different plots?

Lines 352-353: "The likelihood of shoot infection was significantly higher in [trees?] with severe dieback than in those with no dieback". As written, this seems trivial but do you mean that the likelihood of infection of new shoots was significantly higher in trees with more severe dieback?

Line 355: "Shoot mortality in spring was strongly related to leaf necrosis likelihood". Unclear: why 'likelihood of necrosis'? Why not just the amount of necrosis at the end of the previous summer? A similar comments applies to the sentences on lines 357-8 and 358-60.

Line 360: "leaf necrosis did not significantly depend on tree dieback status". This looks like the same point as on lines 342-3 but the statistics are different. Clarification needed.

Line 367: "affected by tree cover". In which direction? (I can see it in Fig. 3 but please say in the text.)

Lines 394-5: "shoot mortality was strongly linked to leaf necrosis likelihood". (a) In which direction? (As for line 367.) (b) Why 'necrosis likelihood'? (As for line 355.)

Line 395: "shoot mortality ... increased with increasing tree cover". This looks like the same point as you made on line 367. Clarification needed.

Lines 415-6: "rachises producing apothecia ... not significantly related to ... leaf necrosis". In Table 4, there's a very large (negative) parameter relating apothecia to necrosis which isn't significant (P=0.3) but a much smaller (positive) parameter relating apothecia to rainfall, which *is* significant (P=0.006). Are you able to give a reason for this apparent disparity?

Discussion

Having written the Results section in a concise manner with little or no interpretation, it would greatly help readers to understand and evaluate the conclusions of your research if you referred to your figures and tables throughout the Discussion. The study is so complex with many explanatory variables which interact to affect three outcome variables differently (necrosis, mortality, apothecia), and have somewhat different effects at many locations, that it's very challenging to follow the Discussion without a clear guide as to which piece of data is relevant.

For example, your very first point in the Discussion, "healthy carriers exist": this requires the reader to piece together results from several tables and figures. There is some explanation in the next sentence but it would help readers greatly if each point in that sentence had references to the relevant data.

A specific point is that your paper is of at least as much interest to plant pathologists as to ecologists but, unlike ecologist, few pathologists are familiar with bayesian statistics. Without more exposition of your conclusions, I frankly believe that most readers will find the Discussion very difficult to follow in detail. I'm familiar with bayesian methods but I myself found it pretty tough.

Looking back at my notes on the Discussion, most of them just say "Explain" - I think that gives you a flavour of what's needed!

Specific points in the Discussion:

Line 438: Do you have any information which would allow you to extrapolate to other regions, beyond France?

Line 446: Likewise, just in French conditions? From observation, I think this is true in the UK because I've very rarely seen apothecia even in early September. If there's hard evidence for this outside France, please give a reference.

Line 492: You say that at low tree cover, crowns are exposed more frequently to adverse temperatures. Could there also be more wind or a dryer micro-environment, both of which are unfavourable to H.frax? (Extrapolating Chumanova's conclusions to the level of individual trees.)

Line 522-3 (sentence about microbiota): Interesting but rather speculative - omit?

Lines 543-5: I don't understand this sentence. I understand that infection can be promoted by rain but necrosis can be suppressed by high temperatures. But why does this explain why rainfall isn't correlated with symptoms, given that high summer rainfall usually involves lower temperatures in summer (but higher in winter)? Or doesn't the correlation between rainfall and lower temperature in summer apply so much in France?

Lines 558-60: "H. fraxineus ability to infect ash leaves and reproduce on the rachises in the forest litter does not appears to limit the pathogen ability to establish throughout France. By contrast, the ability to induce shoot mortality is the limiting step." This is another point I don't understand because a key point of this paper is that reproduction of H.frax on fallen rachises is not closely connected with shoot mortality. Why then do you say that shoot mortality is a limiting step in the establishment of the pathogen? Especially because in Figure 7, you say that shoot mortality is a dead end for the pathogen.

Tables 1 and 2: Please say that positive values indicate an association between the higher levels of the symptom and higher values of the environmental variable.

Figure 2C: On the y-axis, do you mean % shoots with *leaf* necrosis in spring 2015?

Reviewer's Responses to Questions

**Part I - Summary**

Reviewer #1: The study is a multi year field study, bringing together large data sets across multiple sites infected with ash dieback disease. Additionally, they have studied concept of tolerance as opposed to resistance, which to date has been largely assumed rather than tested in the literature.

The manuscript clearly sets out how climatic factors influence leaf necrosis and shoot mortality across France. They clearly set out how prevailing regional climatic conditions will affect the prevalence of ash dieback.

The authors have improved the paper from the first submission and it is now clearer. However, the authors have not corrected all the spelling and grammar mistakes and this must be done to allow for ease of reading. For example, three instances of the mis-spelling 'heathy' were in the introduction, none of which have been corrected in the resubmission. Further examples include line 58- 'This mechanism postulate that some.....' and line 73 '...and it ability to induce damages is an...'. Additionally, Ash is not a proper noun and should be spelt with a small 'a', not a capital 'A' throughout. The paper must be proof read and edited properly before acceptance.

I do not think that the methods are clear enough to allow the reader to fully understand how the authors tested tolerance. I am unsure from the manuscript that the authors have truly identified tolerant trees - as defined by the authors as those trees as having no stem or shoot mortality, but may have leaf necrosis. One of the remarks on response to the first reviewers comments was:

" In many plot, ash dieback was not present since a long time and many individuals healthy the first year anyhow exhibited strong dieback the second."

I do not understand the first part of the sentence- do they mean that many plots were uninfected when they first started sampling but then became infected? More importantly, I am concerned about the second part of the sentence and its implication for the part of the paper suggesting tolerance. If lesions are developing in the second year they are clearly not tolerant.

The authors must be very clear about about their methods for sampling and scoring. Are the same trees rescored where sites are revisited over several years? If a tree is 'healthy' in one year and develops the disease in a subsequent year how is this reflected in the data? From the way the methods are written it appears trees that are scored and sampled from in any one year are classified according to that score. The method states that sites are revisited over 1-6 years but doesn't say if the same trees are revisited and rescored. If they are rescored in a subsequent year how are the data analysed if a tree is healthy one year but exhibited dieback the next? If a tree is tolerant then one wouldn't expect it to have dieback symptoms (shoot mortality or stem lesions) the following year. Stem lesions will often appear over the winter period and not become apparent until the following year. I would suggest that if a tree does form stem lesions it is not tolerant, but from the manuscript I cannot deduce how the authors deal with this. If this can't be clarified and it shown that trees remain tolerant over many years (which would indicate they are truly tolerant), than those claims can not be stated.

Reviewer #2: The authors have addresses all previous concerns adequately and have considerably improved the manuscript which is now in my opinion ready for publication with minor cosmetic changes.

**Part II – Major Issues: Key Experiments Required for Acceptance**

Reviewer #1: The authors must address the issue stated in the summary regarding the sampling procedure with regards to tolerance. The method is not clear enough to discern if they really are investigating tolerance and the reply to reviewers from the first manuscript gives me cause for concern. At the moment, there is not enough evidence provided that the trees are tolerant to be confident of the results presented.

Reviewer #2: /

**Part III – Minor Issues: Editorial and Data Presentation Modifications**

Reviewer #1: The manuscript has spelling and grammar errors throughout which must be corrected. Several of these have not been addressed from the first submission.

Reviewer #2: line 86: replace where with when

L210: "an engine" should probably be replaced by a better term, e.g. logging machinery?

L227: remove: "this is a common procedure frequently used in logistic regression."

Fig 2 could be improved by adding p.values

The quality of all figures in the latest submission seem to be very low (pixelated/unfocused/not sharp) they may need to be generated at higher resolution.

L370: please elaborate on "As no meso-climate effect was included in this model, this was expected." in line 254 you state that "The Env.l [i] and Env.s [i] random factors were taken as proxies for the environment, combining meso/micro climate and plot conditions (host density, Infection history)." both Env factors seem to be in the model (table 1).

373-373: unclear sentence: "with increasing shoot mortality with increasing defoliation level"

380-381: BIC values for "shoot mortality vs climate" model are the same for both time frames. Is that correct?

401-404: improve/clarify English. Use "Moreover, it was not linked to ... " rather than "Moreover, it was

significantly linked neither to ..."

432: "carrier trees without crown dieback symptoms" would be better than "healthy carrier" would be be

436-438: "...it should not be able to induce significant dieback in large part of southern France." I feel the conclusion is a bit too strong given the variation of modelling outcomes in fig S2. More nuanced wording reflecting the effect of parameters presented in Fig S2 is required.

454: change capital "High"  "high" also be more specific and clearly state that high levels of "leaf necrosis" rather that "levels" are not required for rachis colonisation.

455: "this would explain the observed result." feels out of place

459:"within soil reach" better description of the sampling position required.

461: "more humid close to soil and thus more favourable to H. fraxineus infection." (reference missing)

511-523: could this also be due to natural selection of less virulent local strains of the pathogen?

534-538: split in 2 sentences please.

556-557: reword "was by now never reported" to "This is the first report of..."

figure 7 is a great addition but could be improved. Changing red and blue colour (perhaps as a scale) to designate only positive (shades of red) and negative (shades of blue) effects of the studied parameters rather than different parts of disease cycle would convene the gist of the study more clearly. The different parts of the disease cycle could use black and white arrows instead.

Similarly the Fig 7 caption could better describe the actual dynamics of the model and try to avoid ambiguous terms such as "moderate", "presumably" which distract from the point of the figure. I propose the following caption format: "Higher summer rainfall and higher spring temperatures are associated with higher rates of leaf necrosis leading go higher shot necrosis especially in high tree cover. Lower summer temperature .... "

I hope that the authors will soon expand their model/research to predict leaf necrosis/shoot necrosis/reproduction rate of ADB in the future climate scenarios in FR and beyond to shed light on the future of European ash.

PLOS authors have the option to publish the peer review history of their article (what does this mean?). If published, this will include your full peer review and any attached files.

Reviewer #1: No

Reviewer #2: **Yes: **Matevz Papp-Rupar
---

## [Editor Report · Decision Letter 2]

8 Feb 2023

Dear Dr Marçais,

Thank you very much for submitting your manuscript "Ability of the Ash dieback pathogen to reproduce and to induce damage on its host are controlled by different environmental parameters" for consideration at PLOS Pathogens. As with all papers reviewed by the journal, your manuscript was reviewed by members of the editorial board and by several independent reviewers. The reviewers appreciated the attention to an important topic. Based on the reviews, we are likely to accept this manuscript for publication, providing that you modify the manuscript according to the review recommendations.

Thank-you for revising this paper thoroughly in line with my recommendations. Thank-you too for explaining how you've edited the paper and why you haven't accepted certain of my suggestions. It was much easier to follow the Discussion this time, particularly because of the way you've related your conclusions to the tables and figures. I've noted a couple of places where some further cross-reference to figures may be appropriate. As I was going through that paper, I also noted some minor points about language and occasionally terminology (you use 'likelihood' in a couple of figures; as you're presenting Bayesian statistics, I presume you mean [posterior] probability and saying likelihood could be confusing). None of these changes count as major revisions, however.

You make some very interesting points about how the development of the pathogen on the rachis is separate from its pathogenesis on leaves and shoots. This could result in the ultimate outome of the ash dieback epidemic being selection for tolerance rather than resistance, as in Fraxinus mandshurica in Japan.

This has been a very demanding paper to edit and I know the reviewers found it challenging too. However, the journey has been worth it and I look forward to seeing the fully finished version.

Sincerely,

James K.M. Brown

Guest Editor

PLOS Pathogens

Bart Thomma

Section Editor

PLOS Pathogens

Kasturi Haldar

Editor-in-Chief

PLOS Pathogens

orcid.org/0000-0001-5065-158X

Michael Malim

Editor-in-Chief

PLOS Pathogens

orcid.org/0000-0002-7699-2064

Thank-you for revising this paper thoroughly in line with my recommendations. Thank-you too for explaining how you've edited the paper and why you haven't accepted certain of my suggestions. It was much easier to follow the Discussion this time, particularly because of the way you've related your conclusions to the tables and figures. I've noted a couple of places where some further cross-reference to figures may be appropriate. As I was going through that paper, I also noted some minor points about language and occasionally terminology (you use 'likelihood' in a couple of figures; as you're presenting Bayesian statistics, I presume you mean [posterior] probability and saying likelihood could be confusing). None of these changes count as major revisions, however.

You make some very interesting points about how the development of the pathogen on the rachis is separate from its pathogenesis on leaves and shoots. This could result in the ultimate outome of the ash dieback epidemic being selection for tolerance rather than resistance, as in Fraxinus mandshurica in Japan.

This has been a very demanding paper to edit and I know the reviewers found it challenging too. However, the journey has been worth it and I look forward to seeing the fully finished version.

Reviewer Comments (if any, and for reference):

Figure Files:

Data Requirements:

Reproducibility:

References:

---

## [Editor Report · Decision Letter 3]

14 Mar 2023

Dear Dr Marçais,

We are pleased to inform you that your manuscript 'Ability of the Ash dieback pathogen to reproduce and to induce damage on its host are controlled by different environmental parameters' has been provisionally accepted for publication in PLOS Pathogens.

Best regards,

James K.M. Brown

Guest Editor

PLOS Pathogens

Bart Thomma

Section Editor

PLOS Pathogens

Kasturi Haldar

Editor-in-Chief

PLOS Pathogens

orcid.org/0000-0001-5065-158X

Michael Malim

Editor-in-Chief

PLOS Pathogens

orcid.org/0000-0002-7699-2064
---

## [Editor Report · Acceptance letter]

18 Apr 2023

Dear Dr Marçais,

We are delighted to inform you that your manuscript, "Ability of the Ash dieback pathogen to reproduce and to induce damage on its host are controlled by different environmental parameters," has been formally accepted for publication in PLOS Pathogens.

Best regards,

Kasturi Haldar

Editor-in-Chief

PLOS Pathogens

orcid.org/0000-0001-5065-158X

Michael Malim

Editor-in-Chief

PLOS Pathogens

orcid.org/0000-0002-7699-2064